# Recursive Inference for Variational Autoencoders

**Minyoung Kim**[1]
[1]Samsung AI Center
Cambridge, UK
mikim21@gmail.com

**Vladimir Pavlovic**[1,2]
[2]Rutgers University
Piscataway, NJ, USA
vladimir@cs.rutgers.edu

## Abstract

Inference networks of traditional Variational Autoencoders (VAEs) are typically amortized, resulting in relatively inaccurate posterior approximation compared to instance-wise variational optimization. Recent semi-amortized approaches were proposed to address this drawback; however, their iterative gradient update procedures can be computationally demanding. To address these issues, in this paper we introduce an accurate amortized inference algorithm. We propose a novel recursive mixture estimation algorithm for VAEs that iteratively augments the current mixture with new components so as to maximally reduce the divergence between the variational and the true posteriors. Using the functional gradient approach, we devise an intuitive learning criteria for selecting a new mixture component: the new component has to improve the data likelihood (lower bound) and, at the same time, be as divergent from the current mixture distribution as possible, thus increasing representational diversity. Compared to recently proposed boosted variational inference (BVI), our method relies on amortized inference in contrast to BVI's non-amortized single optimization instance. A crucial benefit of our approach is that the inference at test time requires a single feed-forward pass through the mixture inference network, making it significantly faster than the semi-amortized approaches. We show that our approach yields higher test data likelihood than the state-of-the-art on several benchmark datasets.

## 1 Introduction

Accurately modeling complex generative processes for high dimensional data (e.g., images) is a key task in deep learning. In many application fields, the Variational Autoencoder (VAE) [13, 28] was shown to be very effective for this task, endowed with the ability to interpret and directly control the latent variables that correspond to underlying hidden factors in data generation, a critical benefit over synthesis-only models such as GANs [7]. The VAE adopts the *inference network* (aka encoder) that can perform test-time inference using a single feed-forward pass through a neural network. Although this feature, known as *amortized inference*, allows VAE to circumvent otherwise time-consuming procedures of solving the instance-wise variational optimization problem at test time, it often results in inaccurate posterior approximation compared to the instance-wise variational optimization [4].

Recently, semi-amortized approaches have been proposed to address this drawback. The main idea is to use an amortized encoder to produce a reasonable initial iterate, followed by instance-wise posterior fine tuning (e.g., a few gradient steps) to improve the posterior approximation [11, 14, 22, 26]. This is similar to the test-time model adaptation of the MAML [5] in multi-task (meta) learning. However, this iterative gradient update may be computationally expensive during both training and test time: for training, some of the methods require Hessian-vector products for backpropagation, while at test time, one has to perform extra gradient steps for fine-tuning the variational optimization. Moreover, the performance of this approach is often very sensitive to the choice of the gradient step size and the number of gradient updates.

In this paper, we consider a different approach; we build a mixture encoder model, for which we propose a recursive estimation algorithm that iteratively augments the current mixture with a new component encoder so as to reduce the divergence between the resulting variational and the true posteriors. While the outcome is a (conditional) mixture inference model, which could also be estimated by end-to-end gradient descent [33], our recursive estimation method is more effective and less susceptible to issues such as the mixture collapsing. This resiliency is attributed to our specific learning criteria for selecting a new mixture component: the new component has to improve the data likelihood (lower bound) and, at the same time, be as divergent as possible from the current mixture distribution, thus increasing the mixture diversity.

Although a recent family of methods called *Boosted Variational Inference* (BVI) [8, 20, 21, 2, 24] tackles this problem in a seemingly similar manner, our approach differs from BVI in several aspects. Most notably, we address the recursive inference in VAEs in the form of amortized inference, while BVI is developed within the standard VI framework, leading to a non-amortized single optimization instance, inappropriate for VAEs in which the decoder also needs to be simultaneously learned. Furthermore, for the regularization strategy, required in the new component learning stage to avoid degenerate solutions, we employ the *bounded KL loss* instead of the previously used entropy regularization. This approach is better suited for amortized inference network learning in VAEs, more effective as well as numerically more stable than BVI (Sec. 3.1 for detailed discussions).

Another crucial benefit of our approach is that the inference at test time is accomplished using a single feed-forward pass through the mixture inference network, a significantly faster process than the inference in semi-amortized methods. We show that our approach empirically yields higher test data likelihood than standard (amortized) VAE, existing semi-amortized approaches, and even the high-capacity flow-based encoder models on several benchmark datasets.

## 2 Background

We denote by $\mathbf{x}$ observation (e.g., image) that follows the unknown distribution $p_d(\mathbf{x})$. We aim to learn the VAE model that fits the given iid data $\{\mathbf{x}^i\}_{i=1}^N$ sampled from $p_d(\mathbf{x})$. Specifically, letting $\mathbf{z}$ be the underlying latent vector, the VAE is composed of a prior $p(\mathbf{z}) = \mathcal{N}(\mathbf{z}; \mathbf{0}, \mathbf{I})$ and the conditional model $p_{\boldsymbol{\theta}}(\mathbf{x}|\mathbf{z})$ where the latter, also referred to as the *decoder*, is defined as a tractable density (e.g., Gaussian) whose parameters are the outputs of a deep network with weight parameters $\boldsymbol{\theta}$.

To fit the model, we aim to maximize the data log-likelihood, $\sum_{i=1}^N \log p_{\boldsymbol{\theta}}(\mathbf{x}^i)$ where $p_{\boldsymbol{\theta}}(\mathbf{x}) = \mathbb{E}_{p(\mathbf{z})}[p_{\boldsymbol{\theta}}(\mathbf{x}|\mathbf{z})]$. As evaluating the marginal likelihood exactly is infeasible, the variational inference aims to approximate the posterior by a density in some tractable family, that is, $p_{\boldsymbol{\theta}}(\mathbf{z}|\mathbf{x}) \approx q_{\boldsymbol{\lambda}}(\mathbf{z}|\mathbf{x})$ where $q_{\boldsymbol{\lambda}}(\mathbf{z}|\mathbf{x})$ is a tractable density (e.g., Gaussian) with parameters $\boldsymbol{\lambda}$. For instance, if the Gaussian family is adopted, then $q_{\boldsymbol{\lambda}}(\mathbf{z}|\mathbf{x}) = \mathcal{N}(\mathbf{z}; \boldsymbol{\mu}, \boldsymbol{\Sigma})$, where $\{\boldsymbol{\mu}, \boldsymbol{\Sigma}\}$ constitutes $\boldsymbol{\lambda}$. The approximate posterior $q_{\boldsymbol{\lambda}}(\mathbf{z}|\mathbf{x})$ is often called the *encoder*. It is well known that the marginal log-likelihood is lower-bounded by the so-called *evidence lower bound* (ELBO, denoted by $\mathcal{L}$),

$$\log p_{\boldsymbol{\theta}}(\mathbf{x}) \geq \mathcal{L}(\boldsymbol{\lambda}, \boldsymbol{\theta}; \mathbf{x}) := \mathbb{E}_{q_{\boldsymbol{\lambda}}(\mathbf{z}|\mathbf{x})}\big[\log p_{\boldsymbol{\theta}}(\mathbf{x}|\mathbf{z}) + \log p(\mathbf{z}) - \log q_{\boldsymbol{\lambda}}(\mathbf{z}|\mathbf{x})\big], \qquad (1)$$

where the gap in (1) is exactly the posterior approximation error $\mathrm{KL}(q_{\boldsymbol{\lambda}}(\mathbf{z}|\mathbf{x})||p_{\boldsymbol{\theta}}(\mathbf{z}|\mathbf{x}))$.

Hence, maximizing $\mathcal{L}(\boldsymbol{\lambda}, \boldsymbol{\theta}; \mathbf{x})$ with respect to $\boldsymbol{\lambda}$ for the current $\boldsymbol{\theta}$ and the given input instance $\mathbf{x}$, amounts to finding the density in the variational family that best approximates the true posterior $p_{\boldsymbol{\theta}}(\mathbf{z}|\mathbf{x})$. However, notice that the optimum $\boldsymbol{\lambda}$ must be specific to (i.e., dependent on) the input $\mathbf{x}$, and for some other input point $\mathbf{x}'$ one should do the ELBO optimization again to find the optimal encoder parameter $\boldsymbol{\lambda}'$ that approximates the posterior $p_{\boldsymbol{\theta}}(\mathbf{z}|\mathbf{x}')$. The stochastic variational inference (SVI) [9] directly implements this idea, and the approximate posterior inference for a new input point $\mathbf{x}$ in SVI amounts to solving the ELBO optimization on the fly by gradient ascent.

However, the downside is computational overhead since we have to perform iterative gradient ascent to have approximate posterior $q_{\boldsymbol{\lambda}}(\mathbf{z}|\mathbf{x})$ for a new input $\mathbf{x}$. To remedy this issue, one can instead consider an ideal function $\boldsymbol{\lambda}^*(\mathbf{x})$ that maps each input $\mathbf{x}$ to the optimal solution $\arg\max_{\boldsymbol{\lambda}} \mathcal{L}(\boldsymbol{\lambda}, \boldsymbol{\theta}; \mathbf{x})$. We then introduce a deep neural network $\boldsymbol{\lambda}(\mathbf{x}; \boldsymbol{\phi})$ with the weight parameters $\boldsymbol{\phi}$ as a universal function approximator of $\boldsymbol{\lambda}^*(\mathbf{x})$. Then the ELBO, now denoted as $\mathcal{L}(\boldsymbol{\phi}, \boldsymbol{\theta}; \mathbf{x})$, is optimized with respect to $\boldsymbol{\phi}$. This approach, called the *amortized* variational inference (AVI), was proposed in the original VAE [13]. A clear benefit of it is the computational speedup thanks to the feed-forward passing $\boldsymbol{\lambda}(\mathbf{x}; \boldsymbol{\phi})$ used to perform posterior inference for a new input $\mathbf{x}$.

Although AVI is computationally more attractive, it is observed that the quality of data fitting is degraded due to the amortization error, defined as an approximation error originating from the difference between $\boldsymbol{\lambda}^*(\mathbf{x})$ and $\boldsymbol{\lambda}(\mathbf{x}; \boldsymbol{\phi})$ [4]. That is, the AVI's computational advantage comes at the expense of reduced approximation accuracy; the SVI posterior approximation can be more accurate since we minimize the posterior approximation error $\text{KL}(q_{\boldsymbol{\lambda}}(\mathbf{z}|\mathbf{x})||p_{\boldsymbol{\theta}}(\mathbf{z}|\mathbf{x}))$ *individually* for each input $\mathbf{x}$. To address this drawback, the *semi-amortized* variational inference (SAVI) approaches have been proposed in [11, 22, 14]. The main idea is to use the amortized encoder to produce a reasonably good initial iterate for the subsequent SVI optimization. The parameters $\boldsymbol{\phi}$ of the amortized encoder are trained in such a way that several steps of warm-start SVI gradient ascent would yield reduction of the instance-wise posterior approximation error, which is similar in nature to the gradient-based meta learning [5] aimed at fast adaptation of the model to a new task in the multi-task meta learning.

However, the iterative gradient update procedure in SAVI is computationally expensive during both training and test times. For training, it requires backpropagation for the objective that involves gradients, implying the need for Hessian evaluation (albeit finite difference approximation). More critically, at test time, the inference requires a time-consuming gradient ascent optimization. Moreover, its performance is often quite sensitive to the choice of the gradient step size and the number of gradient updates; and it is difficult to tune these parameters to achieve optimal performance-efficiency trade-off. Although more recent work [26] mitigated the issue of choosing the step size by the first-order approximate solution method with the Laplace approximation, such linearization of the deep decoder network restricts its applicability to the models containing only fully connected layers, and makes it difficult to be applied to more structured models such as convolutional networks.

## 3 Recursive Mixture Inference Model (Proposed Method)

Our method is motivated by the premise of the semi-amortized inference (SAVI), i.e., refining the variational posterior to further reduce the difference from the true posterior. However, instead of doing the direct SVI gradient ascent as in SAVI, we introduce another amortized encoder model that augments the first amortized encoder to reduce the posterior approximation error.

Formally, let $q_{\boldsymbol{\phi}}(\mathbf{z}|\mathbf{x})$ be our amortized encoder model[1] with the parameters $\boldsymbol{\phi}$. For the current decoder $\boldsymbol{\theta}$, the posterior approximation error $\text{KL}(q(\mathbf{z}|\mathbf{x})||p_{\boldsymbol{\theta}}(\mathbf{z}|\mathbf{x}))$ equals $-\mathcal{L}(q, \boldsymbol{\theta}; \mathbf{x})$ (up to constant).[2] The goal is to find another amortized encoder model $q'(\mathbf{z}|\mathbf{x})$ with the parameters $\boldsymbol{\phi}'$ such that, when convexly combined with $q(\mathbf{z}|\mathbf{x})$ in a mixture $\epsilon q' + (1 - \epsilon)q$ for some small $\epsilon > 0$, the resulting *reduction of the posterior approximation error*, $\Delta \text{KL} := \mathcal{L}(\epsilon q' + (1 - \epsilon)q, \boldsymbol{\theta}; \mathbf{x}) - \mathcal{L}(q, \boldsymbol{\theta}; \mathbf{x})$, is maximized. That is, we seek $\boldsymbol{\phi}'$ that maximizes $\Delta \text{KL}$.

**Compared to SAVI.** The added encoder $q'$ can be seen as the means for correcting $q$, to reduce the mismatch between $q$ and the true $p_{\boldsymbol{\theta}}(\mathbf{z}|\mathbf{x})$. In SAVI, this correction is done by explicit gradient ascent (finetuning) along $\boldsymbol{\phi}$ for every inference query, at train or test time, which is computationally expensive. In contrast, we learn a differential amortized encoder at training time, which is fixed at test time, requiring only a single neural network feed-forward pass to obtain the approximate posterior.

This encoder correction-by-augmentation can continue by regarding the mixture $\epsilon q' + (1 - \epsilon)q$ as our current inference model to which another new amortized encoder will be added, with the recursion repeated a few times. This leads to a *mixture* model for the encoder, $Q(\mathbf{z}|\mathbf{x}) = \alpha_0 q(\mathbf{z}|\mathbf{x}) + \alpha_1 q'(\mathbf{z}|\mathbf{x}) + \cdots$, where $\sum_m \alpha_m = 1$. The main question is how to find the next encoder model to augment the current mixture $Q$. We do this by the functional gradient approach [6, 23].

**Functional gradients for mixture component search.** Following the functional gradient framework [6, 23], the (ELBO) objective for the mixture $Q(\mathbf{z}|\mathbf{x})$ can be expressed as a *functional*, namely a function that takes a density function $Q$ as input,

$$J(Q) := \mathbb{E}_{Q(\mathbf{z}|\mathbf{x})} \big[ \log p_{\boldsymbol{\theta}}(\mathbf{x}|\mathbf{z}) + \log p(\mathbf{z}) - \log Q(\mathbf{z}|\mathbf{x}) \big]. \tag{2}$$

Let $Q(\mathbf{z}|\mathbf{x})$ be our current mixture. We aim to find $q(\mathbf{z}|\mathbf{x})$ to be added to $Q$ by convex combination,

$$Q(\mathbf{z}|\mathbf{x}) \leftarrow \epsilon q(\mathbf{z}|\mathbf{x}) + (1 - \epsilon)Q(\mathbf{z}|\mathbf{x}) \tag{3}$$

for some small $\epsilon > 0$, that maximizes our objective functional $J$. To this end we take the functional gradient of the objective $J(Q)$ with respect to $Q$. For a given input $\mathbf{x}$, we regard the function $Q(\mathbf{z}|\mathbf{x})$

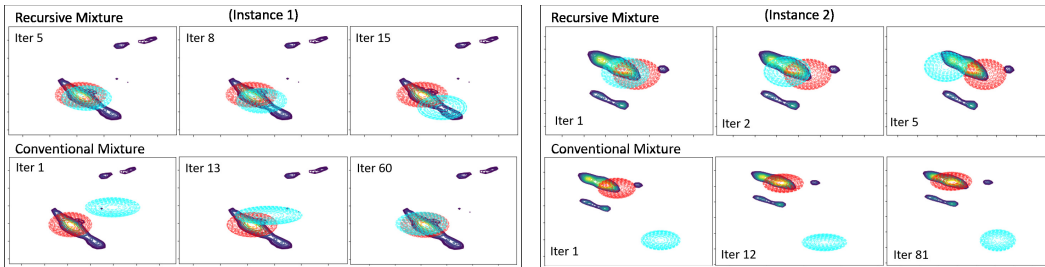

Figure 1: Illustration on MNIST using 2D latent $\mathbf{z}$ space. Results on two data instances (left and right) are shown. (**Top**) Our recursive estimation: The progress of learning the second mixture component is shown from left to right. The contour shows the true posterior $p(\mathbf{z}|\mathbf{x})$, the red is $q_0(\mathbf{z}|\mathbf{x})$, the cyan is the second component that we learn here $q_1(\mathbf{z}|\mathbf{x})$. We only trained $q_1$; remaining parameters (of the decoder and $q_0$) are fixed. Parameters of $q_1$ are initialized to those of $q_0$. (**Bottom**) Conventional (blind) mixture estimation by end-to-end gradient ascent. For the instance 1 (left), the two components collapse onto each other. For the second (right), a single component (red) becomes dominant while the other (cyan) stays away, unutilized, from the support of the true posterior. The cyan is initialized randomly to be different from the red (otherwise, it constitutes a local minimum).

as an infinite-dimensional vector indexed by $\mathbf{z}$, and take the partial derivative at each $\mathbf{z}$, which yields:

$$\frac{\partial J(Q)}{\partial Q(\mathbf{z}|\mathbf{x})} = \log p_{\boldsymbol{\theta}}(\mathbf{x}|\mathbf{z}) + \log p(\mathbf{z}) - \log Q(\mathbf{z}|\mathbf{x}) - 1. \tag{4}$$

Since we have a convex combination (3), the steepest ascent direction (4) needs to be projected onto the feasible function space $\{q(\cdot|\mathbf{x}) - Q(\cdot|\mathbf{x}) : q \in \mathcal{Q}\}$ where $\mathcal{Q} = \{q_{\boldsymbol{\phi}}\}_{\boldsymbol{\phi}}$ is the set of variational densities realizable by the parameters $\boldsymbol{\phi}$. Formally we solve the following optimization:

$$\max_{q \in \mathcal{Q}} \left\langle q(\cdot|\mathbf{x}) - Q(\cdot|\mathbf{x}), \ \frac{\partial J(Q)}{\partial Q(\cdot|\mathbf{x})} \right\rangle, \tag{5}$$

where $\langle \cdot, \cdot \rangle$ denotes the inner product in the function space. Using (4), and considering all training samples $\mathbf{x} \sim p_d(\mathbf{x})$, the optimization (5) can be written as:

$$\max_{\boldsymbol{\phi}} \ \mathbb{E}_{p_d(\mathbf{x})} \Big[ \mathbb{E}_{q_{\boldsymbol{\phi}}(\mathbf{z}|\mathbf{x})} \big[ \log p_{\boldsymbol{\theta}}(\mathbf{x}|\mathbf{z}) + \log p(\mathbf{z}) - \log Q(\mathbf{z}|\mathbf{x}) \big] \Big], \tag{6}$$

where the outer expectation is with respect to the data distribution $p_d(\mathbf{x})$. By adding and subtracting $\log q_{\boldsymbol{\phi}}(\mathbf{z}|\mathbf{x})$ to and from the objective, we see that (6) can be rephrased as follows:

$$\max_{\boldsymbol{\phi}} \ \mathbb{E}_{p_d(\mathbf{x})} \Big[ \mathcal{L}(\boldsymbol{\phi}, \boldsymbol{\theta}; \mathbf{x}) + \mathrm{KL}(q_{\boldsymbol{\phi}}(\mathbf{z}|\mathbf{x}) || Q(\mathbf{z}|\mathbf{x})) \Big]. \tag{7}$$

Note that (7) gives us very intuitive criteria of how the new encoder component $q_{\boldsymbol{\phi}}$ should be selected: it has to maximize the ELBO (the first objective term), and at the same time, $q_{\boldsymbol{\phi}}$ should be *different* from the current mixture $Q$ (the KL term). That is, our next encoder has to keep explaining the data well (by large ELBO) while increasing the diversity of the encoder distribution (by large KL), concentrating on those regions of the latent space that were poorly represented by the current $Q$. This supports our original intuition stated at the beginning of this section. See Fig. 1 for the illustration.

**Why recursive estimation.** Although we eventually form a (conditional) mixture model for the variational encoder, and such a mixture model can be estimated by end-to-end gradient descent, our recursive estimation is efficient and less susceptible to the known issues of blind mixture estimation, including collapsed mixture components and domination by a single component. This resiliency is attributed to our specific learning criteria for selecting a new mixture component: improve the data likelihood and at the same time be as distinct as possible from the current mixture, thus increasing diversity. See Fig. 1 for an illustrative comparison between our recursive and blind mixture estimation.

## 3.1 Optimization Strategy

Although we discussed the key idea of recursive mixture estimation, that is, at each step, fixing the current mixture $Q$ and add a new component $q$, it should be noted that the previously added components $q$'s (and their mixing proportions) need to be refined every time we update the decoder

**Algorithm 1** Recursive Learning Algorithm for Mixture Inference Model.

---

**Input:** Initial $\{q_m(\mathbf{z}|\mathbf{x};\boldsymbol{\phi}_m)\}_{m=0}^M$, $\{\epsilon_m(\mathbf{x};\boldsymbol{\eta}_m)\}_{m=1}^M$, and $p_{\boldsymbol{\theta}}(\mathbf{x}|\mathbf{z})$. Learning rate $\gamma$. KL bound $C$.
**Output:** Learned inference and decoder models.
**Let:** $Q_m = (1-\epsilon_m)Q_{m-1} + \epsilon_m q_m$ $(m = 1 \ldots M)$, $Q_0 = q_0$. $\mathrm{BKL}(p||q) = \max(C, \mathrm{KL}(p||q))$.
**repeat**
    Sample a batch of data $\mathbf{B}$ from $p_d(\mathbf{x})$.
    Update $q_0(\mathbf{z}|\mathbf{x};\boldsymbol{\phi}_0)$: $\boldsymbol{\phi}_0 \leftarrow \boldsymbol{\phi}_0 + \gamma\nabla_{\boldsymbol{\phi}_0}\mathbb{E}_{\mathbf{x}\sim\mathbf{B}}\big[\mathcal{L}(q_0,\boldsymbol{\theta};\mathbf{x})\big]$.
    **for** $m = 1,\ldots,M$ **do**
        Update $q_m(\mathbf{z}|\mathbf{x};\boldsymbol{\phi}_m)$: $\boldsymbol{\phi}_m \leftarrow \boldsymbol{\phi}_m + \gamma\nabla_{\boldsymbol{\phi}_m}\mathbb{E}_{\mathbf{x}\sim\mathbf{B}}\big[\mathcal{L}(q_m,\boldsymbol{\theta};\mathbf{x}) + \mathrm{BKL}(q_m||Q_{m-1})\big]$.
        Update $\epsilon_m(\mathbf{x};\boldsymbol{\eta}_m)$: $\boldsymbol{\eta}_m \leftarrow \boldsymbol{\eta}_m + \gamma\nabla_{\boldsymbol{\eta}_m}\mathbb{E}_{\mathbf{x}\sim\mathbf{B}}\big[\mathcal{L}\big((1-\epsilon_m)Q_{m-1} + \epsilon_m q_m,\boldsymbol{\theta};\mathbf{x}\big)\big]$.
    **end for**
    Update $p_{\boldsymbol{\theta}}(\mathbf{x}|\mathbf{z})$: $\boldsymbol{\theta} \leftarrow \boldsymbol{\theta} + \gamma\nabla_{\boldsymbol{\theta}}\mathbb{E}_{\mathbf{x}\sim\mathbf{B}}\big[\mathcal{L}(Q_M,\boldsymbol{\theta};\mathbf{x})\big]$.
**until** convergence

---

parameters $\boldsymbol{\theta}$. This is due to the VAE framework in which we have to learn the decoder in conjunction with the inference model, one of the main differences from the previous BVI approaches (See Sec. 4).

To this end, we consider a mixture model $Q$ that consists of the *fixed* number ($M$) of components added to the initial component (denoted by $q_0$), namely

$$Q(\mathbf{z}|\mathbf{x}) = \alpha_0(\mathbf{x})q_0(\mathbf{z}|\mathbf{x}) + \sum_{m=1}^M \alpha_m(\mathbf{x})q_m(\mathbf{z}|\mathbf{x}), \qquad (8)$$

where $q_m(\mathbf{z}|\mathbf{x})$ $(m = 0,\ldots,M)$ are all amortized encoders whose parameters are denoted by $\boldsymbol{\phi}_m$, and $\alpha_m$ are the mixing proportions. Since the impact of each component can be different from instance to instance, we consider functions $\alpha_m(\mathbf{x})$, instead of scalars. To respect the idea of recursively adding components (i.e., $q_m$ with $\epsilon_m$), the mixing proportions conform to the following implicit structure:

$$\alpha_m(\mathbf{x}) = \epsilon_m(\mathbf{x}) \prod_{j=m+1}^M (1 - \epsilon_j(\mathbf{x})) \ \text{ for } m = 0, 1, \ldots, M \ \ (\text{let } \epsilon_0(\mathbf{x}) = 1). \qquad (9)$$

This is derived from the recursion, $Q_m = (1-\epsilon_m)Q_{m-1} + \epsilon_m q_m$ for $m = 1,\ldots,M$, where we denote by $Q_m$ the mixture formed by $q_0, q_1, \ldots, q_m$ with $\epsilon_0(= 1)$, $\epsilon_1, \ldots, \epsilon_m$, and $Q_0 := q_0$. Hence $Q_M = Q$. Note also that we model $\epsilon_m(\mathbf{x})$ as neural networks $\epsilon_m(\mathbf{x};\boldsymbol{\eta}_m)$ with parameters $\boldsymbol{\eta}_m$.

Now we describe our recursive mixture learning algorithm. As we seek to update all components simultaneously together with the decoder $\boldsymbol{\theta}$, we employ gradient ascent optimization with *all parameters iteratively and repeatedly*. Our algorithm is described in Alg. 1. Notice that for the $\phi$ update in the algorithm, we used the BKL which stands for *Bounded KL*, in place of KL. The KL term in (7) is to be *maximized*, and it can be easily unbounded; In typical situations, $\mathrm{KL}(q||Q)$ can become arbitrarily large by having $q$ concentrate on the region where $Q$ has zero support. To this end, we impose an upper barrier on the KL term, that is, $\mathrm{BKL}(q||Q) = \max(C, \mathrm{KL}(q||Q))$, so that increasing KL beyond the barrier point $C$ gives no incentive. $C = 500.0$ works well empirically.

Similar degeneracy issues have been dealt with in the previous BVI approaches for non-VAE variational inference [8, 20]. Most approaches attempted to regularize small entropy when optimizing the new components to be added. However, the entropy regularization may be less effective for the iterative refinement of the mixture components within the VAE framework, since we have indirect control of the component models (and their entropy values) only through the density parameter networks $\boldsymbol{\lambda}(\mathbf{x};\phi)$ in $q_{\boldsymbol{\lambda}(\mathbf{x};\phi)}(\mathbf{z}|\mathbf{x})$ (i.e., amortized inference). Furthermore, it encourages the component densities to have large entropy all the time as a side effect, which can lead to a suboptimal solution in certain situations. Our upper barrier method, on the other hand, regularizes the component density only if they are too close (within the range of $C$ KL divergence) to the current mixture, rendering it better chance to find an optimal solution outside the $C$-ball of the current mixture. In fact, the empirical results in Sec. 5.3 demonstrate that our strategy leads to better performance.

The nested loops in Alg. 1 may appear computationally costly, however, the outer loop usually takes a few epochs (usually no more than 20) since we initialize all components $q_m$ identically with the trained encoder parameters of the standard VAE (afterwards, the components quickly move away from each other due to the BKL term). The mixture order $M$ (the number of the inner iterations) is typically small as well (e.g., between 1 and 4), which renders the algorithm fairly efficient in practice.

# 4 Related Work

The VAE's issue of amortization error was raised recently [4], and the semi-amortized inference approaches [11, 22, 14] attempted to address the issue by performing the SVI gradient updates at test time. Alternatively one can enlarge the representational capacity of the encoder network, yet still amortized inference. A popular approach is the flow-based models that apply nonlinear invertible transformations to VAE's variational posterior [30, 12]. The transformations could be complex autoregressive mappings, while they can also model full covariance matrices via efficient parametrization to represent arbitrary rotations, i.e., cross-dimensional dependency. Our use of functional gradient in designing a learning objective stems from the framework in [6, 23]. Mathematically elegant and flexible in the learning criteria, the framework was more recently exploited in [3] to unify seemingly different machine learning paradigms. Several mixture-based approaches aimed to extend the representational capacity of the variational inference model. In [32] the variational parameters were mixed with a flexible distribution. In [31] the prior is modeled as a mixture (aggregate posterior).

**Boosted VI.** Previously, there were approaches to boost the inference network in variational inference similar to our idea [8, 20, 21, 2, 24], where some of them [20, 21, 2] focused on theoretical convergence analysis, inspired by the Frank-Wolfe [10] interpretation of the greedy nature of the algorithm in the infinite-dimensional (function) space. However, these approaches all aimed for stochastic VI in the non-VAE framework, hence non-amortized inference, whereas we consider amortized inference in the VAE framework in which both the decoder and the inference model need to be learned. We briefly summarize the main differences between the previous BVI approaches and ours as follows: 1) We learn $Q(\mathbf{z}|\mathbf{x})$, a density functional of input $\mathbf{x}$, while BVI optimizes $Q(\mathbf{z})$, a single variational density (not a function of $\mathbf{x}$), and thus involves only single optimization. 2) Within the VAE framework, as the decoder is not optimal in the course of training, we update the decoder and all the inference components iteratively and repeatedly. 3) To avoid degeneracy in KL maximization, we employ the bounded KL instead of BVI's entropy penalization, better suited for amortized inference and more effective in practice. 4) The instant impacts of the components, $\epsilon(\mathbf{x})$ are also modeled input-dependent (as neural networks) rather than tunable scalars as in BVI.

# 5 Evaluations

We test the proposed recursive inference model[3] on several benchmark datasets. We highlight improved test likelihood scores and reduced inference time, compared to the semi-amortized VAEs. We also contrast with flow models that aim to increase modeling accuracy using high capacity encoders.

**Competing approaches. VAE**: The standard VAE model (amortized inference) [13, 28]. **SA**: The semi-amortized VAE [11]. We fix the SVI gradient step size as $10^{-3}$, but vary the number of SVI steps from $\{1, 2, 4, 8\}$. **IAF**: The autoregressive-based flow model for the encoder $q(\mathbf{z}|\mathbf{x})$ [12], which has richer expressiveness than VAE's Gaussian encoder. **HF**: The Householder flow encoder model that represents the full covariance using the Householder transformation [30]. The numbers of flows for IAF and HF are chosen from $\{1, 2, 4, 8\}$. **ME**: For a baseline comparison, we also consider the same mixture encoder model, but unlike our recursive mixture learning, the model is trained conventionally, end-to-end; all mixture components' parameters are updated simultaneously. The number of mixture components is chosen from $\{2, 3, 4, 5\}$. **RME**: Our proposed recursive mixture encoder model. We vary the number of additional components $M$ from $\{1, 2, 3, 4\}$, leading to mixture order 2 to 5. All components are initialized identically with the VAE's encoder. See Supplement for the details.

**Datasets. MNIST** [18], **OMNIGLOT** [17], **SVHN** [25], and **CelebA** [19]. We follow train/test partitions provided in the data, where $10\%$ of the training sets are randomly held out for validation. For CelebA, we randomly split data into $80\%/10\%/10\%$ train/validation/test sets.

**Network architectures**. We adopt the convolutional neural networks for the encoder and decoder models for all competing approaches. This is because the convolutional networks are believed to outperform fully connected networks for many tasks in the image domain [16, 29, 27]. We also provide empirical evidence in the Supplement by comparing the test likelihood performance between the two architectures.[4] For the details of the network architectures, refer to the Supplement.

Table 1: Test log-likelihood scores estimated by IWAE sampling. The parentheses next to model names indicate: the number of SVI steps in SA, the number of flows in IAF and HF, and the mixture order in ME and RME. The superscripts are the standard deviations. The best (on average) results are boldfaced in red. In each column, the statistical significance of the difference between the best model (red) and each competing model, is depicted as color: anything non-colored indicates $p \leq 0.01$ (strongly distinguished), $p \in (0.01, 0.05]$ as yellow-orange, $p \in (0.05, 0.1]$ as orange, $p > 0.1$ as red orange (little evidence of difference) by the Wilcoxon signed rank test. Best viewed in color.

| Dataset | MNIST | | OMNIGLOT | | SVHN | | CelebA | |
|---|---|---|---|---|---|---|---|---|
| dim(z) | 20 | 50 | 20 | 50 | 20 | 50 | 20 | 50 |
| VAE | $930.7^{3.9}$ | $1185.7^{3.9}$ | $501.6^{1.6}$ | $801.6^{4.0}$ | $4054.5^{14.3}$ | $5363.7^{21.4}$ | $12116.4^{25.3}$ | $15251.9^{39.7}$ |
| SA$^{(1)}$ | $921.2^{2.3}$ | $1172.1^{1.8}$ | $499.3^{2.5}$ | $792.7^{7.9}$ | $4031.5^{19.0}$ | $5362.1^{35.7}$ | $12091.1^{21.6}$ | $15285.8^{29.4}$ |
| SA$^{(2)}$ | $932.0^{2.4}$ | $1176.3^{3.4}$ | $501.0^{2.7}$ | $793.1^{4.8}$ | $4041.5^{15.5}$ | $5377.0^{23.2}$ | $12087.1^{21.5}$ | $15252.7^{29.0}$ |
| SA$^{(4)}$ | $925.5^{2.6}$ | $1171.3^{3.5}$ | $488.2^{1.8}$ | $794.4^{1.9}$ | $4051.9^{22.2}$ | $5391.7^{20.4}$ | $12116.3^{20.5}$ | $15187.3^{27.9}$ |
| SA$^{(8)}$ | $928.1^{3.9}$ | $1183.2^{3.4}$ | $490.3^{2.8}$ | $799.4^{2.7}$ | $4041.6^{9.5}$ | $5370.8^{18.5}$ | $12100.6^{22.8}$ | $15096.5^{27.2}$ |
| IAF$^{(1)}$ | $934.0^{3.3}$ | $1180.6^{2.7}$ | $489.9^{1.9}$ | $788.8^{4.1}$ | $4050^{9.4}$ | $5368.3^{11.5}$ | $12098.0^{20.6}$ | $15271.2^{28.6}$ |
| IAF$^{(2)}$ | $931.4^{3.7}$ | $1190.1^{1.9}$ | $494.9^{1.4}$ | $795.7^{2.7}$ | $4054.6^{10.5}$ | $5360.0^{10.0}$ | $12104.5^{21.8}$ | $15262.2^{27.8}$ |
| IAF$^{(4)}$ | $926.3^{2.6}$ | $1178.1^{1.6}$ | $496.0^{2.0}$ | $775.1^{2.2}$ | $4048.6^{8.7}$ | $5338.1^{10.2}$ | $12094.6^{22.6}$ | $15261.0^{28.1}$ |
| IAF$^{(8)}$ | $934.1^{2.4}$ | $1150.0^{2.2}$ | $498.8^{2.3}$ | $774.7^{2.9}$ | $4042.0^{9.6}$ | $5341.8^{10.1}$ | $12109.3^{22.0}$ | $15241.5^{27.9}$ |
| HF$^{(1)}$ | $917.2^{2.6}$ | $1204.3^{4.0}$ | $488.6^{2.0}$ | $795.9^{3.3}$ | $4028.8^{9.7}$ | $5372.^{10.1}$ | $12077.2^{31.4}$ | $15240.5^{27.6}$ |
| HF$^{(2)}$ | $923.9^{3.1}$ | $1191.5^{10.8}$ | $495.9^{1.8}$ | $784.5^{4.8}$ | $4030.7^{9.9}$ | $5376.6^{10.2}$ | $12093.0^{25.6}$ | $15258.2^{30.3}$ |
| HF$^{(4)}$ | $927.3^{2.8}$ | $1197.2^{1.5}$ | $487.0^{2.7}$ | $799.7^{3.2}$ | $4038.4^{9.7}$ | $5371.8^{9.8}$ | $12082.0^{27.0}$ | $15266.5^{29.5}$ |
| HF$^{(8)}$ | $928.5^{3.1}$ | $1184.1^{1.8}$ | $488.3^{2.4}$ | $794.6^{4.0}$ | $4035.9^{8.9}$ | $5351.1^{11.1}$ | $12087.3^{25.5}$ | $15248.7^{29.7}$ |
| ME$^{(2)}$ | $926.7^{3.0}$ | $1152.8^{1.7}$ | $491.7^{1.4}$ | $793.4^{3.8}$ | $4037.2^{11.0}$ | $5343.2^{13.1}$ | $12072.7^{23.3}$ | $15290.5^{29.3}$ |
| ME$^{(3)}$ | $933.1^{4.1}$ | $1162.8^{4.7}$ | $491.2^{2.1}$ | $807.5^{4.9}$ | $4053.8^{16.1}$ | $5367.7^{15.8}$ | $12100.3^{21.7}$ | $15294.6^{28.3}$ |
| ME$^{(4)}$ | $914.7^{2.3}$ | $1205.1^{2.3}$ | $491.3^{1.8}$ | $732.0^{3.1}$ | $4061.3^{12.0}$ | $5191.9^{18.5}$ | $12092.2^{22.6}$ | $15270.7^{20.6}$ |
| ME$^{(5)}$ | $920.6^{1.9}$ | $1198.5^{3.5}$ | $478.0^{2.8}$ | $805.7^{3.8}$ | $4057.5^{12.2}$ | $5209.2^{12.8}$ | $12095.3^{25.1}$ | $15268.8^{27.5}$ |
| RME$^{(2)}$ | $943.9^{1.6}$ | $1201.7^{0.9}$ | $508.2^{1.2}$ | $821.0^{3.1}$ | $4085.3^{9.7}$ | $5403.2^{10.2}$ | $12193.1^{23.5}$ | $15363.0^{31.7}$ |
| RME$^{(3)}$ | $945.1^{1.6}$ | $1202.4^{1.0}$ | $507.5^{1.1}$ | $820.4^{0.9}$ | $4085.9^{9.8}$ | $5405.1^{10.4}$ | $12192.3^{23.5}$ | $15365.6^{31.4}$ |
| RME$^{(4)}$ | $945.2^{1.6}$ | $1203.1^{1.0}$ | $509.0^{1.2}$ | $819.9^{0.9}$ | $4080.7^{9.9}$ | $5403.8^{10.2}$ | $12192.6^{23.4}$ | $15364.3^{31.5}$ |
| RME$^{(5)}$ | $945.0^{1.7}$ | $1203.7^{1.0}$ | $509.1^{1.4}$ | $819.9^{0.9}$ | $4086.9^{10.9}$ | $5405.5^{8.5}$ | $12194.2^{11.5}$ | $15366.2^{12.7}$ |

**Experimental setup**. We vary the latent dim(z), small (20) or large (50).[5] To report the test log-likelihood scores $\log p(\mathbf{x})$, we use the importance weighted sampling estimation (IWAE) method [1] with 100 samples (Supplement for details). For each model/dataset, we perform 10 runs with different random train/validation splits, where each run consists of three trainings by starting with different random model parameters, among which only one model with the best validation result is chosen.

### 5.1 Results

The test log-likelihood scores are summarized in Table 1.[6] Overall the results indicate that our recursive mixture encoder (RME) outperforms the competing approaches consistently for all datasets. To see the statistical significance, we performed the one-sided Wilcoxon signed rank test for every pair (the best model, non-best model). The results indicate that this superiority is statistically significant.

**Comparison to ME.** With one exception, specifically ME (4) with dim(z) = 50 on the MNIST, the blind end-to-end mixture learning (ME) consistently underperforms our RME. As also illustrated in Fig. 1, the blind mixture estimation can potentially suffer from mixture collapsing and single dominant component issues. The fact that even the VAE often performs comparably to the ME with different mixture orders supports this observation. On the other hand, our recursive mixture estimation is more robust to the initial parameters. Due to its incremental learning nature, it "knows" the regions in the latent space ill-represented by the current mixture, then updates mixture components to complement those regions. This strategy allows the RME to effectively model highly multi-modal posterior distributions, yielding more robust and accurate variational posterior approximation.

**Comparison to SA.** The semi-amortized approach (SA) sometimes achieves improvement over the VAE, but not consistently. In particular, its performance

Table 2: Test data log-likelihood scores for the **Binary MNIST**. Our results are in the column titled "CNN". The column "FC" is excerpted from [26].

| | CNN | FC |
|---|---|---|
| VAE | -84.49 | -85.38 |
| SA$^{(1)}$ | -83.64 | -85.20 |
| SA$^{(2)}$ | -83.79 | -85.10 |
| SA$^{(4)}$ | -83.85 | -85.43 |
| SA$^{(8)}$ | -84.02 | -85.24 |
| IAF$^{(1)}$ | -83.37 | -84.26 |
| IAF$^{(2)}$ | -83.15 | -84.16 |
| IAF$^{(4)}$ | -83.08 | -84.03 |
| IAF$^{(8)}$ | -83.12 | -83.80 |
| HF$^{(1)}$ | -83.82 | -85.27 |
| HF$^{(2)}$ | -83.70 | -85.31 |
| HF$^{(4)}$ | -83.87 | -85.22 |
| HF$^{(8)}$ | -83.76 | -85.41 |
| ME$^{(2)}$ | -83.77 | - |
| ME$^{(3)}$ | -83.81 | - |
| ME$^{(4)}$ | -83.83 | - |
| ME$^{(5)}$ | -83.75 | - |
| VLAE$^{(2)}$ | - | -83.72 |
| VLAE$^{(3)}$ | - | -83.84 |
| VLAE$^{(4)}$ | - | -83.73 |
| VLAE$^{(5)}$ | - | -83.60 |
| RME$^{(2)}$ | -83.14 | - |
| RME$^{(3)}$ | -83.14 | - |
| RME$^{(4)}$ | -83.09 | - |
| RME$^{(5)}$ | -83.15 | - |

is generally very sensitive to the number of SVI gradient update steps. This is another drawback of the SA, where the gradient-based adaption has to be performed at the test time. Although one could adjust the gradient step size (in place of currently used fixed step size) to improve the performance, there is little principled way to tune the step size at test time that can attain optimal accuracy and inference time trade off. The number of SVI steps in the SA may correspond to the mixture order in our RME model, and the results show that increasing the mixture order usually improves, and not deteriorate, the generalization performance.

**Comparison to IAF/HF.** Although flow models have rich representational capacity, possibly with full covariance matrices (HF), the improvement over the VAE is limited compared to our RME; the models sometimes perform not any better than the VAE. The failure of the flow-based models may originate from the difficulty of optimizing the complex encoder models. (Similar observations were made in related previous work [26]). This result signifies that sophisticated and discriminative learning criteria are critical, beyond just enlarging the structural capacity of the neural networks, similarly observed from the failure of conventional mixtures.

**Non-Gaussian likelihood model.** Our empirical evaluations were predominantly conducted with the convolutional architectures on real-valued image data. For the performance of our model with non-convolutional (fully connected) network architectures, the readers can refer to Table 5 and 6 in the supplementary material. For the binarized input images, we have conducted extra experiments on the **Binary MNIST** dataset. The binary images can be modeled by a Bernoulli likelihood in the decoder. Table 2 summarized the results. We have set the latent dimension $\dim(\mathbf{z}) = 50$, and used the same CNN architectures as before, except that the decoder output is changed from Gaussian to Bernoulli. We also include the reported results from [26] for comparison, which employed the same latent dimension 50 and fully connected encoder/decoder networks with similar model complexity as our CNNs'. As shown, IAF and our RME performs equally the best, although the performance differences among the competing approaches are not very pronounced compared to real-valued image cases.

## 5.2 Test Inference Time

Another key advantage of our recursive mixture inference is the computational efficiency of test-time inference, comparable to that of VAE. Unlike the semi-amortized approaches, where one performs the SVI gradient adaptation at test time, the inference in our RME is merely a single feed forward pass through our mixture encoder network. That is, once training is done, our mixture inference model remains fixed, with no adaptation required.

To verify this empirically, we measure the actual inference time for the competing approaches. The per-batch test inference times (batch size 128) on all benchmark datasets are shown in Tab. 3. To report the results, for each method and each dataset, we run the inference over the entire test set batches, measure the running time, then take the per-batch average. We repeat the procedure five times and report the average. All models are run on the same machine with a single GPU (RTX 2080 Ti), Core i7 3.50GHz CPU, and 128 GB RAM. While we only report test times for $\dim(\mathbf{z}) = 50$, the impact of the latent dimension appears to be less significant.

As expected, the semi-amortized approach suffers from the computational overhead of test-time gradient updates, with the inference time significantly increased as the number of updates increases. Our RME is comparable to VAE, and faster than IAF (with more than a single flow), which verifies our claim. Interestingly, increasing the mixture order in our model

Table 3: Inference time (milliseconds).

|  | MNIST | OMNIG. | SVHN | CELEBA |
|---|---|---|---|---|
| VAE | 3.6 | 4.8 | 2.2 | 2.7 |
| SA$^{(1)}$ | 9.7 | 11.6 | 7.0 | 8.4 |
| SA$^{(2)}$ | 18.1 | 19.2 | 15.5 | 13.8 |
| SA$^{(4)}$ | 32.2 | 34.4 | 30.1 | 27.1 |
| SA$^{(8)}$ | 60.8 | 65.7 | 60.3 | 53.8 |
| IAF$^{(1)}$ | 4.8 | 5.7 | 3.4 | 4.4 |
| IAF$^{(2)}$ | 5.9 | 6.4 | 3.7 | 5.1 |
| IAF$^{(4)}$ | 6.2 | 7.0 | 4.7 | 5.7 |
| IAF$^{(8)}$ | 7.7 | 8.2 | 5.7 | 7.7 |
| RME$^{(2)}$ | 4.7 | 5.4 | 3.2 | 4.2 |
| RME$^{(3)}$ | 4.9 | 5.5 | 3.6 | 4.1 |
| RME$^{(4)}$ | 4.6 | 5.3 | 3.5 | 4.2 |
| RME$^{(5)}$ | 4.8 | 5.6 | 3.3 | 4.8 |

rarely affects the inference time, due to intrinsic parallelization of the feed forward pass through the multiple mixture components networks, leading to inference time as fast as that of VAE.

## 5.3 Comparison with Boosted VI's Entropy Regularization

Recall that our RME adopted the bounded KL (BKL) loss to avoid degeneracy in the component update stages. Previous boosted VI (BVI) approaches employ different regularization, namely penalizing small entropy for the new components. However, such indirect regularization can be

Table 4: Comparison with the BVI's entropy regularization [20]. The same color scheme as Tab. 1.

| Dataset | MNIST | | OMNIGLOT | | SVHN | | CelebA | |
|---|---|---|---|---|---|---|---|---|
| dim($\mathbf{z}$) | 20 | 50 | 20 | 50 | 20 | 50 | 20 | 50 |
| RME$^{(2)}$ | $943.9^{1.6}$ | $1201.7^{0.9}$ | $508.2^{1.2}$ | $821.0^{3.1}$ | $4085.3^{9.7}$ | $5403.2^{10.2}$ | $12193.1^{23.5}$ | $15363.0^{31.7}$ |
| RME$^{(3)}$ | $945.1^{1.6}$ | $1202.4^{1.0}$ | $507.5^{1.1}$ | $820.4^{0.9}$ | $4085.9^{9.8}$ | $5405.1^{10.4}$ | $12192.3^{23.5}$ | $15365.6^{31.4}$ |
| RME$^{(4)}$ | $945.2^{1.6}$ | $1203.1^{1.0}$ | $509.0^{1.2}$ | $819.9^{0.9}$ | $4080.7^{9.9}$ | $5403.8^{10.2}$ | $12192.6^{23.4}$ | $15364.3^{31.5}$ |
| RME$^{(5)}$ | $945.0^{1.7}$ | $1203.7^{1.0}$ | $509.1^{1.4}$ | $819.9^{0.9}$ | $4086.9^{10.9}$ | $5405.5^{8.5}$ | $12194.2^{11.5}$ | $15366.2^{12.7}$ |
| BVI$^{(2)}$ | $939.7^{2.8}$ | $1196.2^{2.8}$ | $507.9^{2.2}$ | $817.1^{3.3}$ | $4077.3^{10.3}$ | $5388.2^{10.2}$ | $12133.5^{25.1}$ | $15206.4^{28.2}$ |
| BVI$^{(3)}$ | $939.5^{2.9}$ | $1191.6^{2.9}$ | $507.3^{2.2}$ | $816.6^{3.4}$ | $4076.6^{10.3}$ | $5384.2^{10.5}$ | $12146.5^{22.4}$ | $15249.5^{28.1}$ |
| BVI$^{(4)}$ | $937.8^{2.9}$ | $1191.6^{2.8}$ | $507.8^{2.3}$ | $816.8^{3.4}$ | $4073.1^{10.2}$ | $5371.1^{10.4}$ | $12127.7^{22.3}$ | $15085.8^{28.4}$ |
| BVI$^{(5)}$ | $931.2^{3.0}$ | $1183.1^{2.9}$ | $508.2^{2.3}$ | $816.4^{3.3}$ | $4071.2^{10.2}$ | $5378.1^{10.1}$ | $12092.3^{22.3}$ | $15052.5^{28.0}$ |

less effective for the iterative refinement of the mixture components within the VAE framework (the second last paragraph of Sec. 3.1). To verify this claim, we test our RME models with the BKL loss replaced by the BVI's entropy regularization. More specifically, following the scheme of [20], we replace our BKL loss by $\nu \cdot \mathbb{E}_{q(\mathbf{z}|\mathbf{x})}[-\log q(\mathbf{z}|\mathbf{x})]$ estimated by Monte Carlo, where $\nu = 1/\sqrt{t+1}$ is the impact that decreases as the training iteration $t$.[7] See Tab. 4 for the results. This empirical result demonstrates that our bounded KL loss consistently yields better performance than entropy regularization. We also observe that our BKL loss leads to numerically more stable solutions: For entropy regularization, we had to reduce the learning rate to the tenth of that of BKL to avoid NaNs.

# 6 Conclusion

In this work we addressed the challenge of improving traditional, amortized inference in VAEs using a mixture of inference networks approach. We demonstrated that this method is both effective in increasing the accuracy of inference and computationally efficient, compared to state-of-the-art semi-amortized inference approaches. This is, in part, due to the effectiveness of the functional recursive mixture learning algorithm we devise and the nature of the inference model, which does not need to be adapted during the test phase. As a consequence, our approach yields higher test data likelihood than the competing approaches on several benchmark datasets, but remains as computationally efficient as the conventional VAE inference. Our recursive model currently requires users to supply the mixture order as an input to the algorithm. In our future work, we aim to investigate principled ways of selecting the mixture order (i.e., model augmentation stopping criteria). We also seek to apply our model to domains with structured data, including sequences (e.g., videos, natural language sentences) and graphs (e.g., molecules, 3D shapes).

# Broader Impact

1. **Who may benefit from this research?** For any individuals, practitioners, organizations, and groups who aim to identify the underlying generative process of the high-dimensional structured data via the variational auto-encoding model framework, this research can be a very useful tool that provides highly accurate solutions generalizable to unseen data.

2. **Who may be put at disadvantage from this research?** Not particularly applicable.

3. **What are the consequences of failure of the system?** Any failure of the system that implements our algorithm would not do any serious harm since the failure can be easily detectable at the validation stage, in which case alternative strategies or internal decisions might be looked for.

4. **Whether the task/method leverages biases in the data?** Our method does not leverage biases in the data.

## Footnotes

[1]This is a shorthand for $q_{\boldsymbol{\lambda}(\mathbf{x};\boldsymbol{\phi})}(\mathbf{z}|\mathbf{x})$. We often drop the subscript and use $q(\mathbf{z}|\mathbf{x})$ for simplicity in notation.

[2]We often abuse the notation, either $\mathcal{L}(\boldsymbol{\phi}, \boldsymbol{\theta}; \mathbf{x})$ or $\mathcal{L}(q, \boldsymbol{\theta}; \mathbf{x})$ interchangeably.

[3]The code is publicly available from `https://github.com/minyoungkim21/recmixvae`

[4]Fully-connected decoder architectures are inferior to the deconvnet when the number of parameters are roughly equal. This is why we exclude comparison with the recent [26], but see Supplement for the results.

[5]The results for dim(z) = 10 and 100, also on the **CIFAR10** dataset [15], are reported in the Supplement.

[6]The MNIST results mismatch those reported in the related work (e.g., [31]). Significantly higher scores. This is because we adopt the Gaussian decoder models, not the binary decoders, for all competing methods.

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
