[Supplementary Material]

# Supplementary Material for:
## Recursive Inference for Variational Autoencoders

**Minyoung Kim**[1]
[1]Samsung AI Center
Cambridge, UK
mikim21@gmail.com

**Vladimir Pavlovic**[1,2]
[2]Rutgers University
Piscataway, NJ, USA
vladimir@cs.rutgers.edu

This supplement consists of the following materials:

## 1 Detailed Experimental Setups

### 1.1 Competing Approaches

The competing approaches are summarized as follows:

- **VAE**: The standard VAE model (amortized inference) [6, 16].
- **SA**: The semi-amortized VAE [4]. We fix the SVI gradient step size as $10^{-3}$, but vary the number of SVI steps from $\{1, 2, 4, 8\}$.
- **IAF**: The autoregressive-based flow model for the encoder $q(\mathbf{z}|\mathbf{x})$ [5], which has richer expressiveness than VAE's post-Gaussian encoder. The number of flows is chosen from $\{1, 2, 4, 8\}$.
- **HF**: The Householder flow encoder model that represents the full covariance using the Householder transformation [18]. The number of flows is chosen from $\{1, 2, 4, 8\}$.
- **ME**: For a baseline comparison, we also consider the same mixture encoder model, but unlike our recursive mixture learning, the model is trained conventionally, end-to-end; all mixture components' parameters are updated simultaneously. The number of mixture components is chosen from $\{2, 3, 4, 5\}$.
- **RME**: Our proposed recursive mixture encoder model. We vary the number of the components to be added $M$ from $\{1, 2, 3, 4\}$, leading to mixture order 2 to 5.

In addition, we test our RME model modified to employ the previous Boosted VI's entropy regularization schemes. More specifically, we replace our bounded KL loss with the two entropy regularization methods as follows:

- **BVI-ER1**: Following [12], we replace our bounded KL loss by $\nu \cdot \mathbb{E}_{q(\mathbf{z}|\mathbf{x})}[-\log q(\mathbf{z}|\mathbf{x})]$ estimated by Monte Carlo, where $\nu = 1/\sqrt{t+1}$ is the impact that decreases as the training iteration $t$.

- **BVI-ER2**: Instead of the Monte Carlo estimation of the entropy, we use [3]'s closed-form Gaussian entropy $\log \det \mathbf{\Sigma}$ where $\mathbf{\Sigma}$ is the (diagonal) covariance of the new component $q(\mathbf{z}|\mathbf{x})$.

## 1.2 Datasets

The following benchmark datasets are used. We randomly hold out $10\%$ of the training data as validation sets, except for **CelebA**.

- **MNIST** [10]: $60,000$ training images and $10,000$ test images where each image is of dimension $(28 \times 28 \times 1)$.
- **OMNIGLOT** [9]: $24,345$ training images and $8,070$ test images where each image is of dimension $(28 \times 28 \times 1)$.
- **CIFAR10** [7]: $50,000$ training images and $10,000$ test images where each image is of dimension $(32 \times 32 \times 3)$.
- **SVHN** [13]: $73,257$ training images and $26,032$ test images where each image is of dimension $(32 \times 32 \times 3)$.
- **CelebA** [11]: $202,599$ tightly cropped face images of size $(64 \times 64 \times 3)$. We randomly split the data into $80\%/10\%/10\%$ train/validation/test sets.

## 1.3 Network Architectures

We adopt the convolutional neural networks for both the encoder and decoder models for all competing approaches. This is because the convolutional networks are believed to outperform fully connected networks for many tasks in the image domain [8, 17, 15]. We also provide empirical evidence in Sec. 3 of this Supplement that the fully-connected decoder architecture is inferior to the deconvnet decoder that we adopted, when the two architectures have roughly equal numbers of parameters. This is why we excluded comparison with the recent Laplacian approximation approach of [14] in the main paper. They use the first-order approximate solver method to obtain the mode of the true posterior, but such linearization of a deep network is only computationally feasible for *fully connected* decoder models. On the other hand, our recursive mixture learning admits arbitrary types of encoder/decoder architectures, which is another advantage. In Sec. 3 of this Supplement we empirically compare the performance between the Laplace approximation [14] and our approach.

For the encoder architecture, we first apply $L$ convolutional layers with $(4 \times 4)$-pixels kernels, followed by two fully-connected layers with hidden layers dimension $h$. For the decoder, the input images first go through two fully connected layers, followed by $L$ deconvolution (*transposed convolution*) layers with $(4 \times 4)$-pixels filters. Here, $L = 3$ for all datasets except CelebA which has $L = 4$. The hidden layer dimension $h = 256$ for MNIST/OMNIGLOT and $h = 512$ for the others. For fair comparison, the same convolutional network architectures are used in all competing methods.

For our recursive mixture RME, all mixture components of the inference model are initialized identically with the VAE's encoder. For the ME (blind end-to-end mixture learning), the first mixture component is initialized with the VAE's encoder while the others are chosen randomly. This is because initializing all components identically would constitute a local maximum of the log-likelihood objective function of the ME, making it unable to update the model further. For the IAF, we follow the inverse autoregressive flow modeling [5] where we use the two-layer MADE [2] (with the number of hidden units 500) as the autoregressiveNN network. The base density, which is transformed to a more complex density by the flow, is initialized with the trained VAE's encoder $q(\mathbf{z}|\mathbf{x})$. For the HF, the latents of the base encoder go through a number of linear transformations, followed by the Householder transformation, where the base encoder is also initialized with the VAE's encoder.

The decoder is modeled as transposed convolutional networks. The network architectures are slightly different across the datasets due to different input image dimensions. We summarize the full network architectures in Tab. 1 (MNIST and OMNIGLOT), Tab. 2 (CIFAR10 and SVHN), and Tab. 3 (CelebA).

In our recursive mixture model, we also need to define the impact function $\epsilon(\mathbf{x})$ for each component. We used a fully connected network $\epsilon(\mathbf{x}; \boldsymbol{\eta})$ with one hidden layer of dimension 10. To prevent a new component from overly taking the mixing proportion, we set an upper bound $\epsilon_{\max}$ on the output of

Table 1: Encoder (i.e., each component in our mixture model) and decoder network architectures for MNIST and OMNIGLOT datasets. In the convolutional and transposed convolutional layers, the paddings are properly adjusted to match the input/output dimensions.

| ENCODER | DECODER |
|---|---|
| INPUT: $(28 \times 28 \times 1)$ | INPUT: $\mathbf{z} \in \mathbb{R}^p$ ($p \in \{10, 20, 50, 100\}$) |
| 32 $(4 \times 4)$ CONV.; STRIDE 2; LEAKYRELU (0.01) | FC. 256; RELU |
| 32 $(4 \times 4)$ CONV.; STRIDE 2; LEAKYRELU (0.01) | FC. $3 \cdot 3 \cdot 64$; RELU |
| 64 $(4 \times 4)$ CONV.; STRIDE 2; LEAKYRELU (0.01) | 32 $(4 \times 4)$ TRANSPOSED CONV.; STRIDE 2; RELU |
| FC. 256; LEAKYRELU (0.01) | 32 $(4 \times 4)$ TRANSPOSED CONV.; STRIDE 2; RELU |
| FC. $2 \times p$ ($p = \text{DIM}(\mathbf{z}) \in \{10, 20, 50, 100\}$) | 1 $(4 \times 4)$ TRANSPOSED CONV.; STRIDE 2 |

Table 2: Encoder and decoder network architectures for CIFAR10 and SVHN datasets.

| ENCODER | DECODER |
|---|---|
| INPUT: $(32 \times 32 \times 3)$ | INPUT: $\mathbf{z} \in \mathbb{R}^p$ ($p \in \{10, 20, 50, 100\}$) |
| 32 $(4 \times 4)$ CONV.; STRIDE 2; LEAKYRELU (0.01) | FC. 512; RELU |
| 32 $(4 \times 4)$ CONV.; STRIDE 2; LEAKYRELU (0.01) | FC. $4 \cdot 4 \cdot 64$; RELU |
| 64 $(4 \times 4)$ CONV.; STRIDE 2; LEAKYRELU (0.01) | 32 $(4 \times 4)$ TRANSPOSED CONV.; STRIDE 2; RELU |
| FC. 512; LEAKYRELU (0.01) | 32 $(4 \times 4)$ TRANSPOSED CONV.; STRIDE 2; RELU |
| FC. $2 \times p$ ($p = \text{DIM}(\mathbf{z}) \in \{10, 20, 50, 100\}$) | 3 $(4 \times 4)$ TRANSPOSED CONV.; STRIDE 2 |

the network. This is done by applying the sigmoid function to the output of $\epsilon(\mathbf{x})$, and multiplication by $\epsilon_{\max}$. For all our experiments $\epsilon_{\max} = 0.1$ worked well.

## 1.4 Experimental Setups

For all optimization, we used the Adam optimizer with batch size 128 and learning rate 0.0005. We run the optimization until 2000 epochs. We vary the latent dimension $\dim(\mathbf{z})$, from $\{10, 20, 50, 100\}$. To report the test log-likelihood scores $\log p(\mathbf{x})$, we use the importance weighted sampling estimation (IWAE) method [1]. More specifically,

$$\text{IWAE} = \log \left( \frac{1}{K} \sum_{i=1}^{K} \frac{p(\mathbf{x}, \mathbf{z}_i)}{q(\mathbf{z}_i | \mathbf{x})} \right), \tag{1}$$

where $\mathbf{z}_1, \ldots, \mathbf{z}_K$ are i.i.d. samples from $q(\mathbf{z}|\mathbf{x})$. It can be shown that IWAE lower bounds $\log p(\mathbf{x})$ and can be arbitrarily close to the target as the number of samples $K$ grows. We use $K = 100$ throughout the experiments.

For each model/dataset, we perform 10 runs with different random train/validation splits, where each run consists of three trainings by starting with different random model parameters, among which only one model with the highest validation performance is chosen. To see the statistical significance of difference between competing models, we also performed the one-sided Wilcoxon signed rank test for every pair, namely (the best model vs. each non-best model), using the 10 log-likelihood scores per model.

## 2 Experimental Results

The test log-likelihood scores are summarized in Tab. 7 (MNIST)[1], Tab. 8 (OMNIGLOT), Tab. 9 (CIFAR10), Tab. 10 (SVHN), and Tab. 11 (CelebA). We also report the performance of the entropy

Table 3: Encoder and decoder network architectures for CelebA dataset.

| ENCODER | DECODER |
|---|---|
| INPUT: $(64 \times 64 \times 3)$ | INPUT: $\mathbf{z} \in \mathbb{R}^p$ $(p \in \{10, 20, 50, 100\})$ |
| 32 $(4 \times 4)$ CONV.; STRIDE 2; LEAKYRELU (0.01) | FC. 512; RELU |
| 32 $(4 \times 4)$ CONV.; STRIDE 2; LEAKYRELU (0.01) | FC. $4 \cdot 4 \cdot 64$; RELU |
| 64 $(4 \times 4)$ CONV.; STRIDE 2; LEAKYRELU (0.01) | 64 $(4 \times 4)$ TRANSPOSED CONV.; STRIDE 2; RELU |
| 64 $(4 \times 4)$ CONV.; STRIDE 2; LEAKYRELU (0.01) | 32 $(4 \times 4)$ TRANSPOSED CONV.; STRIDE 2; RELU |
| FC. 512; LEAKYRELU (0.01) | 32 $(4 \times 4)$ TRANSPOSED CONV.; STRIDE 2; RELU |
| FC. $2 \times p$ $(p = \text{DIM}(\mathbf{z}) \in \{10, 20, 50, 100\})$ | 3 $(4 \times 4)$ TRANSPOSED CONV.; STRIDE 2 |

Table 4: (Per-batch) Test inference time (in milliseconds) with batch size 128. The latent dimension $\dim(\mathbf{z}) = 50$.

| | MNIST | OMNIG. | CIFAR10 | SVHN | CELEBA |
|---|---|---|---|---|---|
| VAE | 3.6 | 4.8 | 3.7 | 2.2 | 2.7 |
| SA (1) | 9.7 | 11.6 | 9.8 | 7.0 | 8.4 |
| SA (2) | 18.1 | 19.2 | 16.8 | 15.5 | 13.8 |
| SA (4) | 32.2 | 34.4 | 27.9 | 30.1 | 27.1 |
| SA (8) | 60.8 | 65.7 | 60.5 | 60.3 | 53.8 |
| IAF (1) | 4.8 | 5.7 | 5.1 | 3.4 | 4.4 |
| IAF (2) | 5.9 | 6.4 | 5.6 | 3.7 | 5.1 |
| IAF (4) | 6.2 | 7.0 | 6.3 | 4.7 | 5.7 |
| IAF (8) | 7.7 | 8.2 | 7.6 | 5.7 | 7.7 |
| RME (2) | 4.7 | 5.4 | 4.9 | 3.2 | 4.2 |
| RME (3) | 4.9 | 5.5 | 5.1 | 3.6 | 4.1 |
| RME (4) | 4.6 | 5.3 | 5.1 | 3.5 | 4.2 |
| RME (5) | 4.8 | 5.6 | 5.1 | 3.3 | 4.8 |

regularization schemes introduced in the previous Boosted VI (BVI) approaches. To this end, in our RME, we replace our bounded KL (BKL) loss with the entropy regularization. More specifically, we consider two entropy regularization schemes – **BVI-ER1**: [12]'s regularization of the negative entropy of $q(\mathbf{z}|\mathbf{x})$ whose impact decreases $\frac{1}{\sqrt{t+1}}$ as a function of training iteration $t$, as suggested. **BVI-ER2**: [3]'s Gaussian entropy based regularization (i.e., penalizing small $\log \det \mathbf{\Sigma}$ where $\mathbf{\Sigma}$ is the (diagonal) covariance matrix of the new component $q(\mathbf{z}|\mathbf{x})$ to be optimized. Overall the results indicate that our recursive mixture encoder (RME) outperforms the competing approaches consistently for all datasets.

## 2.1 Test Inference Time

Another key advantage of our recursive mixture model is the computational efficiency of test-time inference, comparable to that of VAE. Unlike the semi-amortized approaches, where one performs the SVI gradient adaptation at test time, the inference in our RME is merely a single feed forward pass through our mixture encoder network. That is, once training is done, our mixture inference model remains fixed, with no adaptation required.

To verify this, we measure the actual inference time for competing approaches. The per-batch inference times (batch size 128) on all benchmark datasets are shown in Tab. 4. To report the results, for each method and each dataset, we run the inference over the entire test set batches, measure the running time, then take the per-batch average. We repeat the procedure five times and report the average. All models are run on the same machine with a single GPU (RTX 2080 Ti), Core i7 3.50GHz CPU, and 128 GB RAM. We only report test times for the latent dimension $\dim(\mathbf{z}) = 50$ as the impact of the latent dimension appears to be less significant.

As expected, the semi-amortized approach (SA) suffers from the computational overhead of test time gradient updates, with the inference time significantly increased as the number of the updates

Table 5: (Fully connected vs. convolutional decoder networks) Test log-likelihood scores (unit in nat). The figures without parentheses are the scores using the fully connected networks, whereas figures in the parentheses are the scores using the convolutional decoder networks. Both architectures have roughly equal number of the weight parameters. The number of linearization steps in the VLAE is chosen from $\{1, 2, 4, 8\}$.

| | MNIST | | OMNIGLOT | |
|---|---|---|---|---|
| | $\text{DIM}(\mathbf{z}) = 10$ | $\text{DIM}(\mathbf{z}) = 50$ | $\text{DIM}(\mathbf{z}) = 10$ | $\text{DIM}(\mathbf{z}) = 50$ |
| VAE | 563.6 (685.1) | 872.6 (1185.7) | 296.8 (347.0) | 519.4 (801.6) |
| SA (1) | 565.1 (688.1) | 865.8 (1172.1) | 297.6 (344.1) | 489.0 (792.7) |
| SA (2) | 565.3 (682.2) | 868.2 (1176.3) | 295.3 (349.5) | 534.1 (793.1) |
| SA (4) | 565.9 (683.5) | 852.9 (1171.3) | 294.8 (342.1) | 497.8 (794.4) |
| SA (8) | 564.9 (684.6) | 870.9 (1183.2) | 299.0 (344.8) | 500.0 (799.4) |
| VLAE (1) | 590.0 | 922.2 | 307.4 | 644.0 |
| VLAE (2) | 595.1 | 908.8 | 307.6 | 621.4 |
| VLAE (4) | 605.2 | 841.4 | 318.0 | 597.7 |
| VLAE (8) | 605.7 | 779.9 | 316.6 | 553.1 |
| RME (2) | 570.9 (697.2) | 888.1 (1201.7) | 298.4 (349.3) | 524.7 (821.0) |
| RME (3) | 571.9 (698.2) | 888.2 (1202.4) | 298.6 (349.9) | 524.8 (820.4) |
| RME (4) | 571.4 (699.0) | 888.1 (1203.1) | 298.8 (350.7) | 525.3 (819.9) |
| RME (5) | 572.2 (699.4) | 888.0 (1203.7) | 298.8 (351.1) | 526.9 (819.9) |

increases. Our RME is comparable to the VAE, and faster than the IAF (with more than a single flow), which verifies our claim. Interestingly, increasing the mixture order in our model rarely affects the inference time, due to intrinsic parallelization of the feed forward pass through the multiple mixture components networks, leading to inference times as fast as those of the single component model (VAE).

## 3   Comparison with Fully-Connected Decoder Networks

In the main paper we used the convolutional networks for both encoder and decoder models. This is a reasonable architectural choice considering that all the datasets are images. Also it is widely believed that convolutional networks outperform fully connected networks for many tasks in the image domain [8, 17, 15]. However, one can alternatively consider fully connected networks for either the encoder or the decoder, or both. Nevertheless, being equal in the number of model parameters, using both convolutional encoder and decoder networks always outperformed the fully connected counterparts. In this section we empirically verify this by comparing the test likelihood performance between the two architectures. We particularly focus on comparing the two architectures (convolutional vs. fully connected) for the *decoder* model alone, while retaining the convolutional network encoder for both cases.

Using the fully connected decoder network allows us to test the recent Laplacian approximation approach [14] (denoted by **VLAE**), which we excluded from the main paper. They employ a first-order approximation solver to find the mode of the true posterior (i.e., linearizing the decoder function), and compute the Hessian of the log-posterior at the mode to define the (full) covariance matrix. This procedure is computationally feasible only for a fully connected decoder model. We conduct experiments on MNIST and OMNIGLOT datasets where the fully connected decoder network consists of two hidden layers and the hidden layer dimensions are chosen to set the total number of weight parameters roughly equal to the convolutional decoder network used in the main paper.

Tab. 5 summarizes the results. Among the fully connected networks, the VLAE achieves the highest performance. Instead of doing SVI gradient updates as in the SAVI method (SA), the VLAE aims to directly solve for the mode of the true posterior by decoder linearization, leading to more accurate posterior refinement without suffering from the step size issue. Our recursive mixture, with the fully connected decoder networks, still improves the VAE's scores, but the improvement is often less than that of the VLAE. However, when compared to the convnet decoder cases, even the conventional VAE significantly outperforms the VLAE. The best VLAE's scores are significantly lower than VAE's using convolutional decoders. Restricted network architecture of the VLAE is its main drawback.

Table 6: (Fully connected networks as decoders) Per-batch inference time (unit in milliseconds) with batch size 128. The figures without parentheses are the times using the fully connected networks, whereas figures in the parentheses are the times using the convolutional decoder networks.

| | MNIST | | OMNIGLOT | |
|---|---|---|---|---|
| | $\mathrm{DIM}(\mathbf{z}) = 10$ | $\mathrm{DIM}(\mathbf{z}) = 50$ | $\mathrm{DIM}(\mathbf{z}) = 10$ | $\mathrm{DIM}(\mathbf{z}) = 50$ |
| VLAE (1) | 10.1 | 12.9 | 11.2 | 12.1 |
| VLAE (2) | 11.2 | 13.4 | 13.2 | 16.9 |
| VLAE (4) | 14.8 | 17.8 | 15.4 | 18.7 |
| VLAE (8) | 20.7 | 30.8 | 22.1 | 26.4 |
| RME (2) | 5.0 (5.0) | 5.0 (4.7) | 5.4 (6.0) | 5.6 (5.4) |
| RME (3) | 4.9 (5.1) | 4.9 (4.9) | 5.9 (5.7) | 5.4 (5.5) |
| RME (4) | 4.9 (5.0) | 4.9 (4.6) | 6.1 (5.9) | 5.9 (5.3) |
| RME (5) | 5.0 (5.1) | 4.7 (4.8) | 5.8 (6.1) | 5.4 (5.6) |

We also compare the test inference times of our recursive mixture model and the VLAE using the fully connected decoder networks. Note that VLAE is a semi-amortized approach, which needs to solve the Laplace approximation at test time. Thus another drawback of VLAE is the computational overhead of inference, which can be demanding as the number of linearization steps increases. The per-batch inference times (batch size 128) are shown in Tab. 6. For the moderate or large linearization steps (e.g., 4 or 8), the inference takes significantly longer than that of our RME (amortized method).

## 4  Pseudo Codes

The following is the pseudocode for the proposed model. The real full Python/PyTorch code is available in https://github.com/minyoungkim21/recmixvae.

```
#### Hyperparameters ####

batch_size = 128              # input batch size for training
n_epochs = 2000               # number of epochs to train
x_dim = (C=1 x H=28 x W=28)   # input dimension
z_dim = 50                    # latent space dimension
learning_rate = 1e-6          # learning rate for ADAM optimizer

num_comps = 5                 # number of mixture components for encoder
eps_regr_nhl = 1              # number of hidden layers for epsilon regressor
eps_regr_dim = 10             # hidden layer dim for epsilon regressor
eps_min = 0.001               # minimum epsilon
eps_max = 0.1                 # maximum epsilon
kl_max = 500.0                # maximum kl(q_k||Q_{k-1}) allowed in the objective

#### Main class ####

import torch.nn as nn

class RecMixVAE(nn.Module):

    self.M = num_comps-1  # components: 0,1,...,M (the number of comps = M+1)
    self.decoder = ConvDecoder(z_dim, x_dim)  # decoder
    self.prior = DiagonalGaussian(mu=zeros, logvar=zeros)  # prior

    # components of encoder (q_0, q_1, ..., q_M)
    self.comps = nn.ModuleList( [ConvEncoder(z_dim, x_dim) for _ in range(num_comps)] )

    # regressors for impacts of components  (eps_0, eps_1, ..., eps_M); note: eps_0 = 1 (const)
    self.eps_regrs = nn.ModuleList( [Const(1.0)] +
        [ BaseBoundedRegressor( x_dim, eps_min, eps_max, eps_regr_nhl, eps_regr_dim )
          for _ in range(num_comps-1) ] )

    def encoder_upto_kth(self, x, k):
```

```
    '''
    Mixture with components q_0(.|x), q_1(.|x), ..., q_k(.|x) is formed.
    More specifically, eg, for k=2,
       Q_{k=2}(.|x) = alpha_0(x) * q_0(.|x) + alpha_1(x) * q_1(.|x) + alpha_2(x) * q_2(.|x)
    where
       alpha_2(x) = eps_2(x)
       alpha_1(x) = eps_1(x) * (1-eps_2(x))
       alpha_0(x) = eps_0(x) * (1-eps_1(x)) * (1-eps_2(x))
    inputs:
       k = component index (0 <= k <= self.M)
    returns:
       n mixtures for Q_k(.|x) (with k+1 components)
    '''

def encoder_kth_comp(self, x, k):
    '''
    Just return k-th component q_k(.|x)
    inputs:
       k = component index (0 <= k <= self.M)
    returns:
       n distributions (eg, DiagonalGaussian's) q_k(.|x)
    '''
    return self.comps[k](x)[0]

def eval_elbo_for_mixture(self, x, mixture):
    '''
    Evaluate elbo (recon error and kl) for a mixture encoder
    inputs:
       mixture = n mixture distributions from Q(.|x)
    returns:
       ell = E_{Q(z|x)}[ log p(x|z) ]
       kl = KL( Q(z|x) || p(z) )
    '''
    let K = mixture order
    alphas = mixture.logalphas.exp()
    z = samples from q_m(z|x) for m=1...K
    (decoder) evaluate log p(x|z) for z ~ q_m(z|x) for m=1...K
    (prior) evaluate log p(z) for z ~ q_m(z|x) for m=1...K
    evaluate log Q(z|x) for z ~ q_m(z|x) for m=1...K
    return ell = E_{Q(z|x)}[ log p(x|z) ] and kl = KL( Q(z|x) || p(z) )

def forward(self, x, k, loss_type):
    '''
    compute objectives for recursive mixture VAE
    inputs:
       k = component index (0 <= k <= self.M)
       loss_type = either of
            'new_comp': compute elbo(q_k) and kl(q_k||Q_{k-1}) (the latter None if k=0)
            'mixture': compute elbo(Q_k)
    returns:
       loss_type == 'new_comp': elbo(q_k), kl(q_k||Q_{k-1}) (averaged over batch x)
       loss_type == 'mixture': elbo(Q_k) (averaged over batch x)
    '''
  if loss_type == 'new_comp':
       q_z_x = self.encoder_kth_comp(x, k)   # q_k
       Q_z_x = self.encoder_upto_kth(x, k-1) if k>0 else None  # Q_{k-1}
       evaluate elbo(q_k) and kl(q_k||Q_{k-1})
  elif loss_type == 'mixture':
       Q_z_x = self.encoder_upto_kth(x, k)   # Q_k
       ell, kl = self.eval_elbo_for_mixture(x, Q_z_x)
       elbo = ( ell - kl ).mean()

def enable_grad(self, params):
    '''
    Disable the autograd for all parameters except for "params"
```

```
            '''

    #### Main algorithm ####

    model = RecMixVAE()

    while epoch <= n_epochs:

        for batch sampled from the training data:

            # update q_0
            model.enable_grad(model.comps[0])
            elbo, _ = model(batch, 0, loss_type='new_comp')
            update model by backprop with loss = -elbo

            # update (q_m, eps_regr_m) for m=1,...,M
            for m in range(1,model.M+1):

                # update q_m
                model.enable_grad(model.comps[m])
                elbo, kl = model(batch, m, loss_type='new_comp')
                update model by backprop with loss = -elbo + (kl_max - kl).relu()

                # update eps_regr_m
                model.enable_grad(model.eps_regrs[m])
                elbo = model(batch, m, loss_type='mixture')
                update model by backprop with loss = -elbo

            # update decoder
            model.enable_grad(model.decoder)
            elbo = model(batch, model.M, loss_type='mixture')
            update model by backprop with loss = -elbo
```

## Footnotes

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

Table 8: (OMNIGLOT) Test log-likelihood scores (unit in nat). The same interpretation as Tab. 7.

| dim($\mathbf{z}$) | 10 | 20 | 50 | 100 |
|---|---|---|---|---|
| VAE | $347.0^{1.7}$ | $501.6^{1.6}$ | $801.6^{4.0}$ | $917.5^{5.1}$ |
| SA$^{(1)}$ | $344.1^{1.4}$ | $499.3^{2.5}$ | $792.7^{7.9}$ | $905.8^{4.2}$ |
| SA$^{(2)}$ | $349.5^{1.4}$ | $501.0^{2.7}$ | $793.1^{4.8}$ | $920.0^{4.5}$ |
| SA$^{(4)}$ | $342.1^{1.0}$ | $488.2^{1.8}$ | $794.4^{1.9}$ | $914.6^{5.6}$ |
| SA$^{(8)}$ | $344.8^{1.1}$ | $490.3^{2.8}$ | $799.4^{2.7}$ | $942.2^{5.2}$ |
| IAF$^{(1)}$ | $347.8^{1.6}$ | $489.9^{1.9}$ | $788.8^{4.1}$ | $937.4^{7.2}$ |
| IAF$^{(2)}$ | $344.2^{1.6}$ | $494.9^{1.4}$ | $795.7^{2.7}$ | $934.6^{7.3}$ |
| IAF$^{(4)}$ | $347.9^{1.9}$ | $496.0^{2.0}$ | $775.1^{2.2}$ | $920.9^{4.1}$ |
| IAF$^{(8)}$ | $343.9^{1.4}$ | $498.8^{2.3}$ | $774.7^{2.9}$ | $885.7^{2.8}$ |
| HF$^{(1)}$ | $335.5^{1.2}$ | $488.6^{2.0}$ | $795.9^{3.3}$ | $917.0^{2.4}$ |
| HF$^{(2)}$ | $340.6^{1.3}$ | $495.9^{1.8}$ | $784.5^{4.8}$ | $929.4^{3.7}$ |
| HF$^{(4)}$ | $343.3^{1.2}$ | $487.0^{2.7}$ | $799.7^{3.2}$ | $877.5^{4.7}$ |
| HF$^{(8)}$ | $343.3^{1.3}$ | $488.3^{2.4}$ | $794.6^{4.0}$ | $889.2^{4.7}$ |
| ME$^{(2)}$ | $344.2^{1.5}$ | $491.7^{1.4}$ | $793.4^{3.8}$ | $880.3^{3.6}$ |
| ME$^{(3)}$ | $350.3^{1.8}$ | $491.2^{2.1}$ | $807.5^{4.9}$ | $875.9^{4.6}$ |
| ME$^{(4)}$ | $337.7^{1.1}$ | $491.3^{1.8}$ | $732.0^{3.1}$ | $939.8^{8.6}$ |
| ME$^{(5)}$ | $343.0^{1.4}$ | $478.0^{2.8}$ | $805.7^{3.8}$ | $861.9^{7.0}$ |
| RME$^{(2)}$ | $349.3^{1.5}$ | $508.2^{1.2}$ | $\mathbf{821.0}^{3.1}$ | $941.5^{1.7}$ |
| RME$^{(3)}$ | $349.9^{1.6}$ | $507.5^{1.1}$ | $820.4^{0.9}$ | $\mathbf{944.6}^{5.1}$ |
| RME$^{(4)}$ | $350.7^{1.7}$ | $509.0^{1.2}$ | $819.9^{0.9}$ | $944.4^{1.7}$ |
| RME$^{(5)}$ | $\mathbf{351.1}^{1.7}$ | $\mathbf{509.1}^{1.4}$ | $819.9^{0.9}$ | $944.0^{1.6}$ |
| BVI-ER1$^{(2)}$ | $349.2^{1.9}$ | $507.9^{2.2}$ | $817.1^{3.3}$ | $937.9^{5.1}$ |
| BVI-ER1$^{(3)}$ | $350.0^{1.9}$ | $507.8^{2.2}$ | $816.6^{3.4}$ | $936.2^{5.1}$ |
| BVI-ER1$^{(4)}$ | $350.7^{1.5}$ | $507.8^{2.3}$ | $816.8^{3.4}$ | $935.6^{3.8}$ |
| BVI-ER1$^{(5)}$ | $351.1^{1.5}$ | $508.2^{2.3}$ | $816.4^{3.3}$ | $935.7^{3.8}$ |
| BVI-ER2$^{(2)}$ | $349.3^{1.9}$ | $507.8^{2.2}$ | $817.1^{3.4}$ | $937.6^{5.1}$ |
| BVI-ER2$^{(3)}$ | $349.8^{1.9}$ | $507.8^{2.2}$ | $816.6^{3.4}$ | $936.1^{5.1}$ |
| BVI-ER2$^{(4)}$ | $350.7^{1.5}$ | $507.8^{2.2}$ | $816.9^{3.4}$ | $935.6^{3.8}$ |
| BVI-ER2$^{(5)}$ | $351.0^{1.5}$ | $508.1^{2.2}$ | $816.4^{3.4}$ | $935.7^{3.8}$ |

Table 9: (CIFAR10) Test log-likelihood scores (unit in nat). The same interpretation as Tab. 7.

| dim($\mathbf{z}$) | 10 | 20 | 50 | 100 |
|---|---|---|---|---|
| VAE | $1645.7^{4.9}$ | $2089.7^{5.8}$ | $2769.9^{7.1}$ | $3381.0^{14.7}$ |
| SA$^{(1)}$ | $1645.0^{5.6}$ | $2086.0^{6.2}$ | $2765.0^{7.1}$ | $3378.7^{10.4}$ |
| SA$^{(2)}$ | $1648.6^{4.8}$ | $2088.2^{6.6}$ | $2764.1^{7.7}$ | $3377.8^{9.8}$ |
| SA$^{(4)}$ | $1648.5^{5.2}$ | $2083.9^{8.4}$ | $2766.7^{6.6}$ | $3380.2^{7.9}$ |
| SA$^{(8)}$ | $1642.1^{5.4}$ | $2086.0^{6.1}$ | $2766.6^{7.5}$ | $3376.6^{10.6}$ |
| IAF$^{(1)}$ | $1646.0^{4.9}$ | $2081.1^{5.4}$ | $2762.6^{7.2}$ | $3383.7^{7.1}$ |
| IAF$^{(2)}$ | $1642.0^{4.9}$ | $2084.6^{5.6}$ | $2763.0^{4.3}$ | $3373.3^{14.2}$ |
| IAF$^{(4)}$ | $1646.0^{5.1}$ | $2083.2^{6.1}$ | $2760.6^{7.0}$ | $3371.1^{8.1}$ |
| IAF$^{(8)}$ | $1643.6^{4.6}$ | $2087.1^{4.6}$ | $2761.8^{6.9}$ | $3364.0^{9.6}$ |
| HF$^{(1)}$ | $1644.5^{4.4}$ | $2079.1^{5.5}$ | $2757.9^{4.4}$ | $3393.4^{4.7}$ |
| HF$^{(2)}$ | $1636.7^{4.9}$ | $2086.0^{5.9}$ | $2764.7^{4.4}$ | $3384.8^{4.7}$ |
| HF$^{(4)}$ | $1642.1^{4.9}$ | $2082.3^{7.3}$ | $2763.4^{4.4}$ | $3385.5^{4.4}$ |
| HF$^{(8)}$ | $1639.9^{5.4}$ | $2084.7^{6.1}$ | $2765.5^{7.2}$ | $3382.5^{4.3}$ |
| ME$^{(2)}$ | $1643.6^{5.1}$ | $2086.6^{6.8}$ | $2767.9^{9.4}$ | $3378.5^{9.1}$ |
| ME$^{(3)}$ | $1638.6^{5.8}$ | $2079.8^{5.9}$ | $2770.2^{7.8}$ | $3388.1^{7.7}$ |
| ME$^{(4)}$ | $1641.8^{5.4}$ | $2084.7^{6.9}$ | $2763.5^{9.3}$ | $3384.6^{10.3}$ |
| ME$^{(5)}$ | $1641.7^{5.6}$ | $2080.2^{5.9}$ | $2766.1^{6.3}$ | $3351.3^{11.0}$ |
| RME$^{(2)}$ | $1652.3^{5.0}$ | $2095.7^{5.8}$ | $2779.6^{6.6}$ | $3403.0^{6.9}$ |
| RME$^{(3)}$ | $1654.2^{4.9}$ | $\mathbf{2099.1}^{7.2}$ | $\mathbf{2783.0}^{6.1}$ | $3404.2^{6.8}$ |
| RME$^{(4)}$ | $\mathbf{1655.0}^{6.4}$ | $2096.6^{5.9}$ | $2781.1^{6.6}$ | $3403.2^{6.1}$ |
| RME$^{(5)}$ | $1654.5^{4.6}$ | $2098.4^{5.8}$ | $2782.9^{6.4}$ | $\mathbf{3404.6}^{5.7}$ |
| BVI-ER1$^{(2)}$ | $1648.6^{5.1}$ | $2094.4^{5.7}$ | $2775.9^{6.4}$ | $3393.1^{6.8}$ |
| BVI-ER1$^{(3)}$ | $1648.9^{5.0}$ | $2094.7^{5.9}$ | $2776.2^{6.6}$ | $3393.8^{6.5}$ |
| BVI-ER1$^{(4)}$ | $1649.0^{5.1}$ | $2095.0^{5.8}$ | $2776.5^{6.3}$ | $3394.2^{6.6}$ |
| BVI-ER1$^{(5)}$ | $1649.1^{5.2}$ | $2095.1^{5.8}$ | $2776.8^{6.5}$ | $3394.2^{7.7}$ |
| BVI-ER2$^{(2)}$ | $1648.6^{5.1}$ | $2094.4^{5.7}$ | $2775.8^{6.8}$ | $3393.1^{6.6}$ |
| BVI-ER2$^{(3)}$ | $1648.9^{5.0}$ | $2094.7^{5.7}$ | $2776.2^{6.6}$ | $3393.8^{6.5}$ |
| BVI-ER2$^{(4)}$ | $1649.0^{5.1}$ | $2095.0^{5.8}$ | $2776.5^{6.3}$ | $3394.2^{6.2}$ |
| BVI-ER2$^{(5)}$ | $1649.1^{5.1}$ | $2095.1^{5.8}$ | $2776.8^{6.5}$ | $3394.1^{6.1}$ |

Table 10: (SVHN) Test log-likelihood scores (unit in nat). The same interpretation as Tab. 7.

| dim($\mathbf{z}$) | 10 | 20 | 50 | 100 |
|---|---|---|---|---|
| VAE | $3360.2^{9.1}$ | $4054.5^{14.3}$ | $5363.7^{21.4}$ | $6703.0^{28.4}$ |
| SA$^{(1)}$ | $3358.7^{8.9}$ | $4031.5^{19.0}$ | $5362.1^{35.7}$ | $6707.6^{24.8}$ |
| SA$^{(2)}$ | $3356.0^{8.8}$ | $4041.5^{15.5}$ | $5377.0^{23.2}$ | $6697.0^{35.5}$ |
| SA$^{(4)}$ | $3327.8^{8.2}$ | $4051.9^{22.2}$ | $5391.7^{20.4}$ | $6645.1^{19.8}$ |
| SA$^{(8)}$ | $3352.8^{11.5}$ | $4041.6^{9.5}$ | $5370.8^{18.5}$ | $6674.5^{20.9}$ |
| IAF$^{(1)}$ | $3377.1^{8.4}$ | $4050.0^{9.4}$ | $5368.3^{11.5}$ | $6650.3^{15.7}$ |
| IAF$^{(2)}$ | $3362.3^{8.9}$ | $4054.6^{10.5}$ | $5360.0^{10.0}$ | $6671.5^{16.8}$ |
| IAF$^{(4)}$ | $3346.1^{8.7}$ | $4048.6^{8.7}$ | $5338.1^{10.2}$ | $6630.0^{17.2}$ |
| IAF$^{(8)}$ | $3372.6^{8.3}$ | $4042.0^{9.6}$ | $5341.8^{10.1}$ | $6602.0^{10.8}$ |
| HF$^{(1)}$ | $3381.4^{8.9}$ | $4028.8^{9.7}$ | $5372.0^{10.1}$ | $6678.8^{8.8}$ |
| HF$^{(2)}$ | $3342.4^{8.3}$ | $4030.7^{9.9}$ | $5376.6^{10.2}$ | $6672.0^{9.6}$ |
| HF$^{(4)}$ | $3370.0^{8.2}$ | $4038.4^{9.7}$ | $5371.8^{9.8}$ | $6655.2^{9.5}$ |
| HF$^{(8)}$ | $3343.8^{8.2}$ | $4035.9^{8.9}$ | $5351.1^{11.1}$ | $6642.4^{16.5}$ |
| ME$^{(2)}$ | $3352.3^{9.9}$ | $4037.2^{11.0}$ | $5343.2^{13.1}$ | $6670.2^{46.5}$ |
| ME$^{(3)}$ | $3335.2^{10.9}$ | $4053.8^{16.1}$ | $5367.7^{15.8}$ | $6605.6^{9.4}$ |
| ME$^{(4)}$ | $3358.2^{14.9}$ | $4061.3^{12.0}$ | $5191.9^{18.5}$ | $6605.7^{9.2}$ |
| ME$^{(5)}$ | $3360.6^{7.8}$ | $4057.5^{12.2}$ | $5209.2^{12.8}$ | $6604.0^{16.6}$ |
| RME$^{(2)}$ | $3390.0^{8.1}$ | $4085.3^{9.7}$ | $5403.2^{10.2}$ | $\mathbf{6784.7}^{25.0}$ |
| RME$^{(3)}$ | $\mathbf{3392.0}^{12.6}$ | $4085.9^{9.8}$ | $5405.1^{10.4}$ | $6782.7^{9.3}$ |
| RME$^{(4)}$ | $3388.6^{8.3}$ | $4080.7^{9.9}$ | $5403.8^{10.2}$ | $6780.2^{9.4}$ |
| RME$^{(5)}$ | $3391.9^{8.2}$ | $\mathbf{4086.9}^{10.9}$ | $\mathbf{5405.5}^{8.5}$ | $6781.8^{10.0}$ |
| BVI-ER1$^{(2)}$ | $3379.9^{8.2}$ | $4077.3^{10.3}$ | $5388.2^{10.2}$ | $6753.5^{10.0}$ |
| BVI-ER1$^{(3)}$ | $3380.9^{8.1}$ | $4076.6^{10.3}$ | $5384.2^{10.5}$ | $6750.3^{10.6}$ |
| BVI-ER1$^{(4)}$ | $3384.4^{8.1}$ | $4073.1^{10.2}$ | $5371.1^{10.4}$ | $6748.9^{11.3}$ |
| BVI-ER1$^{(5)}$ | $3382.2^{8.4}$ | $4071.2^{10.2}$ | $5378.1^{10.1}$ | $6733.6^{15.3}$ |
| BVI-ER2$^{(2)}$ | $3379.8^{8.1}$ | $4077.3^{9.8}$ | $5388.3^{10.1}$ | $6753.2^{10.1}$ |
| BVI-ER2$^{(3)}$ | $3380.9^{8.4}$ | $4076.7^{9.6}$ | $5383.9^{10.2}$ | $6749.7^{10.7}$ |
| BVI-ER2$^{(4)}$ | $3384.3^{8.2}$ | $4073.2^{9.2}$ | $5371.3^{10.4}$ | $6749.1^{11.1}$ |
| BVI-ER2$^{(5)}$ | $3382.1^{8.4}$ | $4071.2^{10.4}$ | $5377.7^{10.2}$ | $6733.8^{15.0}$ |

Table 11: (CelebA) Test log-likelihood scores (unit in nat). The same interpretation as Tab. 7.

| dim($\mathbf{z}$) | 10 | 20 | 50 | 100 |
|---|---|---|---|---|
| VAE | $9767.7^{36.0}$ | $12116.4^{25.3}$ | $15251.9^{39.7}$ | $17395.5^{32.4}$ |
| SA$^{(1)}$ | $9735.2^{21.4}$ | $12091.1^{21.6}$ | $15285.8^{29.4}$ | $17432.4^{30.4}$ |
| SA$^{(2)}$ | $9754.2^{20.4}$ | $12087.1^{21.5}$ | $15252.7^{29.0}$ | $17434.0^{29.8}$ |
| SA$^{(4)}$ | $9769.1^{20.6}$ | $12116.3^{20.5}$ | $15187.3^{27.9}$ | $17360.5^{28.9}$ |
| SA$^{(8)}$ | $9744.8^{19.4}$ | $12100.6^{22.8}$ | $15096.5^{27.2}$ | $17409.7^{28.0}$ |
| IAF$^{(1)}$ | $9750.3^{27.4}$ | $12098.0^{20.6}$ | $15271.2^{28.6}$ | $17446.4^{30.3}$ |
| IAF$^{(2)}$ | $9794.4^{23.3}$ | $12104.5^{21.8}$ | $15262.2^{27.8}$ | $17449.5^{31.8}$ |
| IAF$^{(4)}$ | $9764.7^{29.5}$ | $12094.6^{22.6}$ | $15261.0^{28.1}$ | $17416.8^{29.8}$ |
| IAF$^{(8)}$ | $9764.0^{21.6}$ | $12109.3^{22.0}$ | $15241.5^{27.9}$ | $17452.5^{39.5}$ |
| HF$^{(1)}$ | $9748.3^{29.5}$ | $12077.2^{31.4}$ | $15240.5^{27.6}$ | $17461.6^{29.9}$ |
| HF$^{(2)}$ | $9765.8^{25.6}$ | $12093.0^{25.6}$ | $15258.2^{30.3}$ | $17479.8^{30.0}$ |
| HF$^{(4)}$ | $9754.3^{23.8}$ | $12082.0^{27.0}$ | $15266.5^{29.5}$ | $17532.7^{30.6}$ |
| HF$^{(8)}$ | $9737.5^{24.5}$ | $12087.3^{25.5}$ | $15248.7^{29.7}$ | $17663.4^{28.7}$ |
| ME$^{(2)}$ | <span style="color:orange">$9825.3^{20.7}$</span> | $12072.7^{23.3}$ | $15290.5^{29.3}$ | $17419.3^{28.7}$ |
| ME$^{(3)}$ | $9797.6^{22.3}$ | $12100.3^{21.7}$ | $15294.6^{28.3}$ | $17395.3^{28.9}$ |
| ME$^{(4)}$ | <span style="color:orange">$9834.9^{25.4}$</span> | $12092.2^{22.6}$ | $15270.7^{20.6}$ | $17458.5^{36.8}$ |
| ME$^{(5)}$ | $9717.0^{23.2}$ | $12095.3^{25.1}$ | $15268.8^{27.5}$ | $17406.8^{31.8}$ |
| RME$^{(2)}$ | <span style="color:orange">$9837.9^{24.6}$</span> | <span style="color:orange">$12193.1^{23.5}$</span> | <span style="color:orange">$15363.0^{31.7}$</span> | <span style="color:orange">$17873.5^{32.8}$</span> |
| RME$^{(3)}$ | <span style="color:orange">$9838.5^{25.0}$</span> | <span style="color:orange">$12192.3^{23.5}$</span> | <span style="color:orange">$15365.6^{31.4}$</span> | <span style="color:orange">$17874.4^{31.2}$</span> |
| RME$^{(4)}$ | <span style="color:red">**$9849.5^{12.1}$**</span> | <span style="color:orange">$12192.6^{23.4}$</span> | <span style="color:orange">$15364.3^{31.5}$</span> | <span style="color:red">**$17875.1^{14.2}$**</span> |
| RME$^{(5)}$ | <span style="color:orange">$9843.5^{25.0}$</span> | <span style="color:red">**$12194.2^{11.5}$**</span> | <span style="color:red">**$15366.2^{12.7}$**</span> | <span style="color:orange">$17874.3^{32.5}$</span> |
| BVI-ER1$^{(2)}$ | $9801.6^{26.1}$ | $12133.5^{25.1}$ | $15206.4^{28.2}$ | $17716.9^{70.3}$ |
| BVI-ER1$^{(3)}$ | $9805.6^{25.7}$ | $12146.5^{22.4}$ | $15249.5^{28.1}$ | $17558.6^{120.1}$ |
| BVI-ER1$^{(4)}$ | $9805.2^{29.3}$ | $12127.7^{22.3}$ | $15085.8^{28.4}$ | $17256.1^{283.9}$ |
| BVI-ER1$^{(5)}$ | $9810.1^{30.7}$ | $12092.3^{22.3}$ | $15052.5^{28.0}$ | $17069.9^{391.8}$ |
| BVI-ER2$^{(2)}$ | $9801.5^{25.3}$ | $12133.6^{28.7}$ | $15207.3^{52.4}$ | $17716.6^{92.1}$ |
| BVI-ER2$^{(3)}$ | $9805.7^{24.9}$ | $12146.6^{25.5}$ | $15249.6^{54.6}$ | $17560.7^{109.2}$ |
| BVI-ER2$^{(4)}$ | $9805.1^{26.3}$ | $12128.7^{34.0}$ | $15084.9^{42.5}$ | $17260.6^{228.6}$ |
| BVI-ER2$^{(5)}$ | $9810.4^{27.8}$ | $12087.5^{48.9}$ | $15051.7^{43.5}$ | $17077.1^{387.6}$ |