[Reviews · NeurIPS 2020]

Review 1

Summary and Contributions: This paper proposes an amortized variational inference technique which uses a mixture distribution. Each component of the mixture is output by a separate neural network encoder, trained to optimize the improvement in the ELBO. The authors compare their technique with semi-amortized VI, normalizing flows, and a non-recursive mixture encoder across a variety of image datasets, where their technique performs well. In terms of inference time, their technique is also faster than semi-amortized VI. Update: After reading the authors' response, the other reviews, and the reviewer discussion, I am increasing my score toward acceptance. My primary concerns were regarding the overstated claims on semi-amortized VI and the evaluation. The authors reasonably addressed these issues in their response, and I am hopeful that they will include these changes (and others) in the final submission. This paper presents and reasonably novel contribution which is also demonstrated reasonably well. As such, I feel that this paper would be a useful addition to the conference.

Strengths: Soundness: The theoretical grounding and empirical evaluation are largely sound. The authors derive their technique in Section 3 and show how this naturally results in an objective for the current component that trades off between the ELBO and a KL from the current approximate posterior distribution. The authors compare their approach against a comprehensive set of baselines, including a standard VAE, semi-amortized VI (which also involves additional computation), normalizing flows (which uses a more expressive distribution), and a non-recursive mixture distribution (which has the same form of distribution). This comparison is performed across a range of image datasets and multiple model sizes using multiple runs. The authors also compare their approach with boosted VI, showing that the KL, rather than entropy, is useful for mixture estimation. Significance: The proposed approach outperforms more expressive VI techniques, particularly IAF. Normalizing flows, such as IAF, are seen as a general purpose technique for improving the flexibility of approximate posterior distributions. If this technique can be further scaled up, it could provide an alternative to normalizing flows, instead based on mixture distributions. This could be significant in improving VI in more complex settings and models. Further, in the authors’ formulation, estimating the components can be parallelized at test time, enabling more expressive distributions with constant time cost. Novelty: While mixture distributions are a common feature of probabilistic models, it seems that the exact formulation of splitting the ELBO as a sequence of mixture components is a novel aspect of this work. It’s perhaps possible that this estimation technique could also be extended to aspects of the generative model or other probabilistic modeling settings. Relevance: This paper focuses on improving variational inference in deep latent variable models, i.e. VAEs. Because these topics relate to deep learning and probabilistic modeling, this paper is relevant to the generative modeling community and the NeurIPS community more broadly.

Weaknesses: Soundness: I did not completely follow the derivation/explanation in Section 3, and I imagine that the definition of Q (with or without the current mixture) could trip up readers. To better understand the method, it would be helpful to re-write and/or expand the method, perhaps in the appendix. Some of the claims around semi-amortized VI are exaggerated, and it may not be entirely fair to frame this work in comparison with semi-amortized VI. In the introduction and background, the authors lump a number of different works under the name “semi-amortized variational inference,” stating that it is difficult to tune the step-size and that it requires estimating Hessians. One of these works, iterative amortized inference (Marino et al., 2018), does not require tuning a step-size or estimating a Hessian, whereas another previous work (Krishnan et al., 2018) only requires tuning the step size. Further, these works still use Gaussian approximate posteriors, focusing on decreasing the amortization gap (Cremer et al., 2018). This paper, in contrast, improves the form of the distribution, which can be seen as decreasing the approximation gap. The authors show, surprisingly, that their method outperforms normalizing flows across a range of datasets. The authors claim that these models tend to overfit. However, from what I can tell, the authors do not provide the training performance of these models. This would be helpful to allow readers to assess these claims. Further, this claim may be overly general; the authors explore a particular family of single latent level convolutional model architectures. It’s unclear whether their technique would equally apply to more complex hierarchical models. The authors utilize multiple inference networks in their approach. While they compare with a non-recursive version of this model, it should be emphasized that their approach contains more parameters than many of the baselines. Significance: Ultimately, the experiments demonstrate slight improvements in performance for a particular family of convolutional architectures on image datasets. To further evaluate and demonstrate the significance of these results, it would be helpful to expand the experiments to include other models and data modalities. One simple additional example would be to use binarized MNIST with a Bernoulli conditional likelihood. This is a fairly standard benchmark in the deep generative modeling community. Novelty: The proposed technique is similar in spirit to boosted VI, and ultimately comes down to a new way to train mixture distributions. While the authors demonstrate the benefits of their particular approach, the difference appears to be truly in this relative training scheme and clamped KL term. In terms of novelty, these are not large contributions. Relevance: No weaknesses.

Correctness: In the experiments, log-densities are reported (in nats). Typically, on image datasets log-probabilities are reported (in bits/dimension). Correcting this would help to improve the ability of readers and other researchers to compare these results with past/future papers. As noted above, some of the claims around semi-amortized VI are overstated or lumped across multiple different works. It should be noted that the authors’ method also requires multiple gradient passes during training. An analysis of the training time of each method would be helpful to assess the additional cost of this approach. Conventionally, many previous works use 5,000 importance-weighted samples to evaluate the marginal log-likelihood. The paper reports the value obtained with 100 samples. I don’t imagine that the results will change significantly, but it would be useful to eventually re-perform these evaluations with additional samples. Also as noted above, the test results for models with normalizing flows are, in many cases, below those for simpler models. The authors claim that this is due to overfitting, but do not provide the training results.

Clarity: Yes, the paper is well written. Section 3 could be slightly improved, particularly to distinguish between the mixture distribution before and after the addition of the current mixture component. The setup for Figure 1 does not appear to be fully explained in the paper. Some details regarding the experiment setup appear to be omitted from the paper and appendix. For instance, from the code, it’s clear that the authors use Gaussian conditional likelihoods, but I could not find this detail in the paper.

Relation to Prior Work: For the most part, the relation to previous work is discussed well. For readers, such as myself, who are unfamiliar with boosted VI, it would be helpful to explain in math how the objective derived here relates to the entropy objective employed in the boosted VI setting. The authors hint at this but it could be further expanded formally. Also, it would be useful to explain further why amortization cannot/has not been applied in this setting thus far. Again, as noted above, the authors lump multiple works together under the name “semi-amortized VI.” The authors should distinguish between the differences in these works, as some of the claims are incorrect/overstated. The authors may also want to cite Iterative refinement of the approximate posterior for directed belief networks by Hjelm et al., 2016 as well as Recurrent inference machines for solving inference problems by Putzky et al., 2017.

Reproducibility: Yes

Additional Feedback: It would be helpful to better distinguish between the amortization gap vs. the approximation gap (Cremer et al., 2018). Much of this paper is framed in terms of comparing with semi-amortized VI, however, the proposed approach uses a more expressive distribution, so this framing is somewhat flawed. Line 51: I would consider changing the word “inappropriate” here, or at least explaining this point in further detail. Line 119: is epsilon in [0,1]? Figure 1: This figure could be improved, e.g. by adding labels to the axes, labeling the mixture components with a legend, etc. It’s also unclear why the cyan component in the conventional mixture in instance 2 does not find a mode of the distribution. Section 3.1: This form of recursive estimation is not explicitly conditioned on the previous components of the distribution, but must instead learn these implicitly through amortization. While this allows parallelized inference at test time, it may be computationally limited. The authors may consider explicitly conditioning on the previous mixture distribution parameters, or, for example, using a recurrent network as the output of the encoder to sequentially generate each mixture component for all dimensions. Section 5: The authors should state clearly that their method contains more parameters in the inference network than most of the baseline methods.


Review 2

Summary and Contributions: The paper proposes a new fashion of constructing encoders in the VAE framework. The idea is to build a mixture of encoders in a recursive manner. The paper is rather clearly written, and all concepts are explained.

Strengths: + The proposed idea is interesting. + The proposed approach is easy to implement and seems to work in practice. + The paper is well-written.

Weaknesses: - All results reported in the paper suggest that all datasets were modeled by Gaussian decoders. This is not the best possible choice for images that are discrete in nature. I wonder whether the same conclusions could be drawn if discretized logistic distributions are used. - The authors claim that the posterior collapse is not an issue. I understand that and buy this argument. However, it would be beneficial to provide empirical evidence for that.

Correctness: The derivations seem to be correct. The reasoning is logical and sound. The only issue I have right now is the choice of the distribution. It seems the authors used Gaussian decoders that is not necessarily a correct choice for image data.

Clarity: The paper is well-written. I do not have any concerns about the clarity of the paper.

Relation to Prior Work: The prior work is properly discussed.

Reproducibility: Yes

Additional Feedback: o I wonder how stable is the proposed procedure. The authors use an additional quantity C, i.e., the barrier point. I imagine if C is too large, then the method could return strange results. I am curious what is the experience of the authors with this matter. ===AFTER THE REBUTTAL=== I would like to thank the authors for their rebuttal. After a discussion with the other reviewers I decided to raise my score to 7. My main concern about the paper is the presentation of the idea. Please try to work on that. But otherwise, good job!


Review 3

Summary and Contributions: After author rebuttal: Thank you very much for the detailed rebuttal. I in particular appreciated the experiments on binary versions of datasets and encourage the authors to include them in the final version of the paper. I also encourage the authors to reconsider/revamp the presentation of the proposed approach (and discussion of related work) given R1's excellent comments. Finally, as noted by R5 it would be good to investigate how the decoupling the inference network/generative model changes the training/optimization dynamics. ========= This paper proposes a method to improve variational approximations to the posterior in the context of learning deep generative models (i.e., variational autoencoders). The approach utilizes richer variational distributions than the standard Gaussian by successfully learning mixture distributions via iterative refinement. This recursive inference approach is found to outperform a variety of existing methods that also increase the flexibility of the posterior.

Strengths: - Well-motivated method that proposes an amortized alternative to methods such as boosted variational inference. Solid contribution to the existing line of work on improving variational approximations within the VAE framework. - Comprehensive comparison against a wide range of existing approaches (including semi-amortized approaches, which still assume a Gaussian variational posterior, and flow-based methods, which increase the flexibility of the variational family)

Weaknesses: - The model is only compared against internally-implemented baselines. In particular, the difference in evaluation setup makes it hard to compare against external baselines. For example, why not apply the approach on the standard binarized setup for MNIST/Omniglot? (It is of course not necessary to achieve "SOTA" results, but it seems unideal to be in a completely different setup to majority of existing approaches) - It would have been interesting to see the effectiveness of this approach as a function of the encoder/decoder capacity. For example, what happens with an autoregressive decoder like the PixelCNN? Can you get away with less powerful encoders in future refinement steps? - The model is only tested on image domains. It would have been interesting to apply this on sequential data such as text (not a huge drawback though).

Correctness: Yes

Clarity: The paper is very clear, well-motivated, and easy to understand

Relation to Prior Work: Yes.

Reproducibility: Yes

Additional Feedback: - Small point: "For training, it requires backpropagation for the objective that involves gradients, implying the need for Hessian evaluation" ==> it would be more accurate to say that we need the Hessian vector product (vs. the full Hessian). - I would encourage the authors to discuss the potential impacts of deep generative models (such as VAEs) in their broader impacts statement


Review 4

Summary and Contributions: A method is proposed for variational autoencoders for better posterior approximation. The method (RME) is like boosting in that mixture components are added iteratively to the approximate posterior, but previously added components are trained together. The mixture coefficients depend on the data point and are trained concurrently with the parameters mixture components. The main contribution is bringing Boosted VI to amortized inference. UPDATE: I've read the authors' response and I stand by my assessment and confidence score.

Strengths: Suboptimal inference in VAEs is an important problem. The paper proposes a novel method in the context of amortized variational inference and demonstrates good experimental results. The figures with the 2D latent space illustrate the strength of the algorithm.

Weaknesses: My main concern is that although the 2D examples look intuitively appealing, they are produced with a given p() (i.e. a fixed decoder). VAEs train q() and p() jointly and so does RME. The interaction between an already hard mixture estimation problem (the encoder mixture) and learning the decoder p() is not explored. - Are multi-modal posteriors needed in VAEs? - Is p(x|z) implicitly regularized by RME to have higher variance? This implicit regularization concern would be alleviated by a more careful experimental setup, which included multiple regularization methods on p() and more effort to tune hyperparameters such as the learning rate and network sizes. Also related, RME(1) is not equivalent to the VAE due to how it updates the encoder and decoder parameters in a coordinate descent manner. Did you do experiments with it? Finally, since the method can be viewed as a way to avoid component collapse in mixture encoders, the Vamprior is a very obvious missing baseline.

Correctness: As alluded to in weaknesses, the experiments could be improved.

Clarity: Yes.

Relation to Prior Work: Estimating mixtures and component collapse has a large literature and should be mentioned in related works. Lines 152-158 are the closest to even mentioning this.

Reproducibility: Yes

Additional Feedback:

[Author Response · NeurIPS 2020]

We are very grateful to all reviewers for their detailed, insightful, and constructive comments and questions. We find many of the comments truly helpful to improve the quality of the paper, and some of them actually enlightened us, correcting some of our initial claims that turn out to be wrong.

Due to lack of time, we apologize for not responding to all questions, and not conducting all extra empirical evaluations suggested by the reviewers. But we believe that they are very important, and we will pursue them in our ongoing study.

Our responses (blue) to reviewers' comments/questions (***black/bold/italic***) are as follows.

***1. Claims on semi-amortized variational inference (SAVI) methods:***

We agree that we made somewhat exaggerated claims on the drawbacks of the SAVI methods. Specifically, some SAVI methods do not suffer from either step size adjustment or Hessian evaluation. We will refine our claims, and also refer to these SAVI methods.

***2. About our claim that the normalizing flow models (eg, IAF) tend to overfit:***

It turns out that it was our faulty claim. As the reviewers (esp., R1) suggested, we checked the training data likelihood scores of the IAF model, and found that oftentimes the training performance was even worse than the vanilla VAE. This implies that the failure of the normalizing flow models may not be because of overfitting, but difficulty of optimizing the complex encoder model. This is also argued/stated in the related work [VLAE19] (the second paragraph in Sec. 5.1).

***3. The increased number of parameters in the proposed model:***

In the paper, we only emphasized the merits of our model without mentioning the drawbacks. Clearly, our model has more parameters to estimate than the SAVI methods. The increased training time (due to the loop over the mixture components for each iteration) is yet another weak point. We will summarize and state the drawbacks of our model in the revised version.

***4. Performance of the proposed model with a non-Gaussian (eg, Bernoulli) decoder, or binarized input images:***

Our empirical evaluations were predominantly conducted with the convolutional architectures on real-valued image data. Several reviewers wondered how the proposed model would perform on different data/network setups. For the performance of our model with non-convolutional (fully connected) network architectures, please refer to Table 5 and 6 in our supplementary material at the time of paper submission.

For the binarized input images, we have conducted extra experiments on the **Binary MNIST** dataset. Please see Table 1 (on the right) for the results. We have set the latent dimension $\dim(\mathbf{z}) = 50$, and used the same CNN architectures as in our paper, except that the decoder output is changed from Gaussian to Bernoulli. We also include the reported results from [VLAE19] for comparison, which employed the same latent dimension 50 and fully connected encoder/decoder networks with similar model complexity as our CNNs'. Due to lack of time, we only report mean scores averaged over three runs. As shown, IAF and our RME performs equally the best, although the performance differences among the competing approaches are not very pronounced compared to real-valued image cases.

***5. The KL bound parameter $C$:***

We haven't tested extensively the impact of $C$. We once tried a few different values $\{100, 500, 1000\}$, and checked that the performance did not vary significantly. Although we guess that having too large or too small value of $C$ would deteriorate the performance, we should do more rigorous empirical study in our ongoing work.

***6. About RME(1). Due to the coordinate descent learning, it is not equivalent to the VAE:***

Although RME(1), the single component mixture model, is optimized with the same loss function, its coordinate descent learning may yield a local optimum different from that of the VAE. We haven't compared the two models specifically, but we believe it is worth testing it empirically.

***7. The VampPrior model as a baseline:***

We agree that VampPrior can be another reasonable baseline in the sense that the model can potentially avoid component collapsing. Although VampPrior adopts a mixture-type *prior* while ours adopts a mixture *encoder*, understanding the relationship between the two, either empirically or theoretically, must be an intriguing research problem.

**References**

[VLAE19] "Variational Laplace Autoencoders", Park et al., ICML 2019

Table 1: Test data log-likelihood scores for the **Binary MNIST**. Our results are in the column titled "CNN". The column "FC" is excerpted from [VLAE19] (Table 2).

| | CNN | FC |
|---|---|---|
| VAE | -84.49 | -85.38 |
| SA(1) | -83.64 | -85.20 |
| SA(2) | -83.79 | -85.10 |
| SA(4) | -83.85 | -85.43 |
| SA(8) | -84.02 | -85.24 |
| IAF(1) | -83.37 | -84.26 |
| IAF(2) | -83.15 | -84.16 |
| IAF(4) | -83.08 | -84.03 |
| IAF(8) | -83.12 | -83.80 |
| HF(1) | -83.82 | -85.27 |
| HF(2) | -83.70 | -85.31 |
| HF(4) | -83.87 | -85.22 |
| HF(8) | -83.76 | -85.41 |
| ME(2) | -83.77 | - |
| ME(3) | -83.81 | - |
| ME(4) | -83.83 | - |
| ME(5) | -83.75 | - |
| VLAE(2) | - | -83.72 |
| VLAE(3) | - | -83.84 |
| VLAE(4) | - | -83.73 |
| VLAE(5) | - | -83.60 |
| RME(2) | -83.14 | - |
| RME(3) | -83.14 | - |
| RME(4) | -83.09 | - |
| RME(5) | -83.15 | - |

[Meta-Review · NeurIPS 2020]

The paper introduces RMIM -- an amortized version of boosted VI (BVI), based on a slightly different objective, which uses a recursively parameterized mixture as the variational posterior. Improving inference in VAEs is an area of wide interest and potentially high impact. The reviewers thought the paper was mostly well written and the approach sensible. The experimental results are encouraging, with RMIM outperforming quite a few baselines, though there were some potential issues with the experimental setup as explained below. The paper has substantial room for improvement however, with the reviewers making several good suggestions. For example, the derivation of the objective needs to be made clearer, including an explanation of how exactly it differs from the derivation in the BVI paper. It would also be good to explore the effect of the parameter C that determines how different the components of the mixture, as it seems crucial to the method. The use of a Gaussian likelihood without dequantization on (at least partially) binary (MNIST and Omniglot) or quantized data (SVHN & CelebA) means that the results cannot be compared to those in the literature and the correctness of baselines cannot be verified. Moreover, the quality of the posterior approximation will be considerably more important when using a Gaussian likelihood (without dequantization) relative to a less sensitive likelihood such as Bernoulli. As a result, the relative performance of the proposed method might be considerably less impressive in a more careful experimental setup. The results on binary MNIST with a Bernoulli likelihood reported in the rebuttal are consistent with this theory. The authors are urged to rerun the experiments on all the datasets using more appropriate likelihoods and include the results in the final version of the paper.